# Collective anti-predator escape manoeuvres through optimal attack and avoidance strategies
Palina Bartashevich [1,2] ✉, James E. Herbert-Read [3,4], Matthew J. Hansen[5], Félicie Dhellemmes [2,5], Paolo Domenici [6], Jens Krause [2,5,7] & Pawel Romanczuk [1,2]

The collective dynamics of self-organised systems emerge from the decision rules agents use to respond to each other and to external forces. This is evident in groups of animals under attack from predators, where understanding collective escape patterns requires evaluating the risks and rewards associated with particular social rules, prey escape behaviour, and predator attack strategies. Here, we find that the emergence of the 'fountain effect', a common collective pattern observed when animal groups evade predators, is the outcome of rules designed to maximise individual survival chances given predator hunting decisions. Using drone-based empirical observations of schooling sardine prey (*Sardinops sagax caerulea*) attacked by striped marlin (*Kajikia audax*), we first find the majority of attacks produce fountain effects, with the dynamics of these escapes dependent on the predator's attack direction. Then, using a spatially-explicit agent-based model of predator-prey dynamics, we show that fountain manoeuvres can emerge from combining an optimal individual prey escape angle with social interactions. The escape rule appears to prioritise maximising the distance to the predator and creates conflict in the effectiveness of predators' attacks and the prey's avoidance, explaining the empirically observed predators' attack strategies and the fountain evasions produced by prey. Overall, we identify the proximate and ultimate explanations for fountain effects and more generally highlight that the collective patterns of self-organised predatory-prey systems can be understood by considering both social escape rules and attack strategies.

High risk and high uncertainty situations require fast and robust reactions by individuals. In such scenarios, it is beneficial for individuals to rely on simple decision heuristics[1–3]. In groups, the interplay of these individual responses spreading through social interactions, and the influence of external environmental driving factors, results in the emergence of complex self-organised collective response patterns with non-trivial consequences for collective and individual performance. This is clearly manifested in the context of predator-prey dynamics in groups. For example, collective patterns such as "vacuoles", "splits", and "waves" are often produced when predators attack grouping prey[4–7]. These collective patterns are thought to be the outcome of prey attempting to flee in ways that maximise their survival chances, and predators attempting to attack groups in ways that facilitate

their hunting success. Surprisingly little is known, however, about why particular social escape rules are adopted by grouping prey to escape predators, the collective escape patterns that are produced by those rules, and whether these rules are robust to different attack scenarios.

One collective anti-predator response is the so-called "fountain effect", whereby prey break into two subgroups, turning in an arched trajectory around an attacking predator and rejoining at its tail, visually appearing like a fountain (see Fig. 1A). This collective response appears to allow slower-moving prey to outmanoeuvre faster but less manoeuvrable predators[8,9], while also allowing the separated subgroups to rejoin after the attack, retaining the benefits of belonging to a larger group. Fountain effects are currently understood to be a by-product of individual escape manoeuvres,

[1]Institute for Theoretical Biology, Department of Biology, Humboldt-Universität zu Berlin, Berlin, Germany. [2]Research Cluster of Excellence "Science of Intelligence", Technische Universität Berlin, Berlin, Germany. [3]Department of Zoology, University of Cambridge, Cambridge, UK. [4]Aquatic Ecology Unit, Department of Biology, University of Lund, Lund, Sweden. [5]Department of Fish Biology, Fisheries and Aquaculture, Leibniz Institute of Freshwater Ecology and Inland Fisheries, Berlin, Germany. [6]IBF-CNR, Consiglio Nazionale delle Ricerche, Area di Ricerca San Cataldo, Via G. Moruzzi No. 1, Pisa, 56124, Italy. [7]Faculty of Life Science, Humboldt-Universität zu Berlin, Berlin, Germany. ✉e-mail: bartashevich.palina@gmx.de

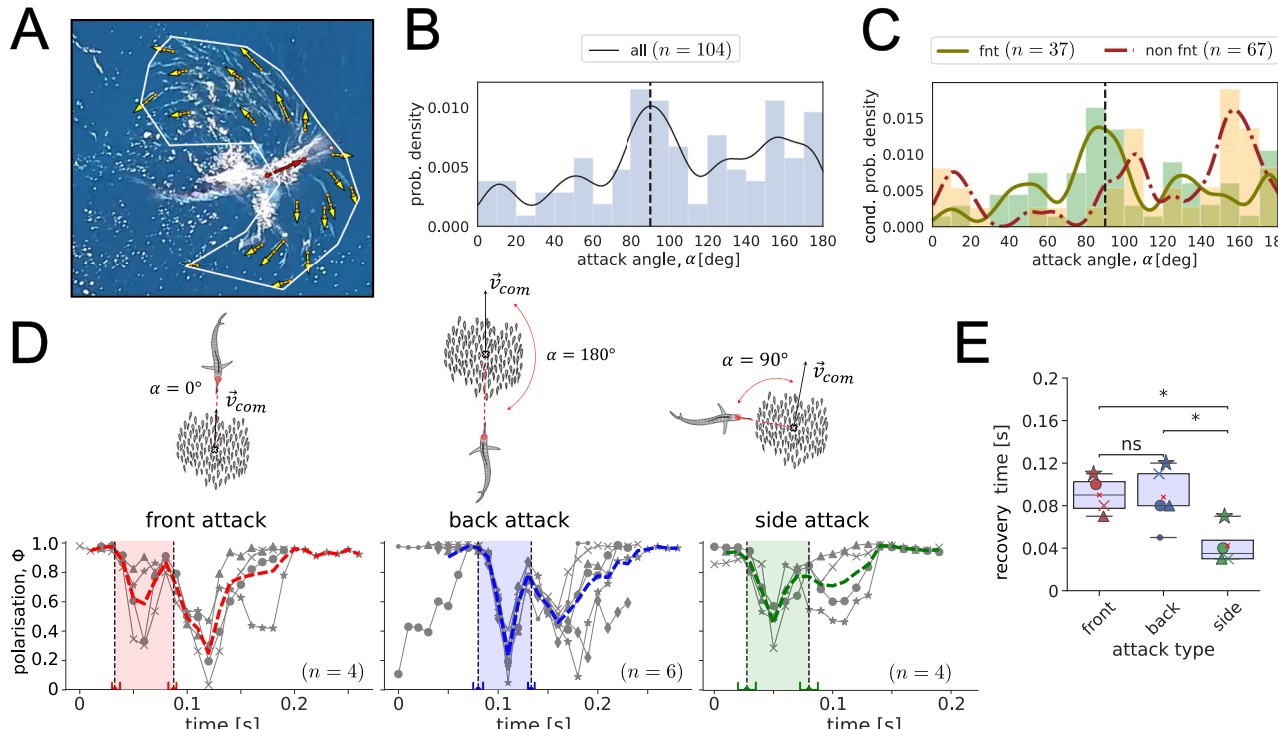

**Fig. 1 | "Fountain effect" in the wild. A** An instance of an annotated snapshot from the aerial footage over an open ocean, capturing the "fountain effect" performed by a school of sardines attacked by a striped marlin. Yellow and red vectors (defined by two dots as the head and the 'tail' on the fish) depict prey and predator velocity orientations, respectively. The arrows' length is independent of the speed. The white polygon describes the borders of the school. The pink dot indicates the predator's bill tip. **B** Probability density distribution of the predator attack directions $\alpha$ estimated between the predator's velocity orientation and the heading direction of the prey's school centre of mass over $n = 104$ recorded attacks. **C** Conditional probability density distribution of the predator attack directions based on the pattern of the observed anti-predator response, "fountain" (in green) or "non fountain" (in red). Each distribution is independently normalised according to the respective number of events ("fnt": $n = 67$ and "non fnt": $n = 37$), resulting in the rescaling of the bar heights (y-axis) compared to the entire distribution in (**B**). Histograms' overlap results in a third colour other than green (for "fnt") and orange (for "non fnt"). **D** Prey polarisation $\Phi$ over time, estimated with 20 randomly annotated sardines as shown in (**A**). Polarisation during each encounter is indicated by a grey line, while the thick line in colour shows the averaged $\Phi$ over the encounters of the respective attack type (14 attacks by 11 marlins). The shaded areas indicate the fountain window times (i.e., "start" and "end" times) averaged over each single encounter (grey line) and therefore not necessarily aligning with the peaks of the averaged polarisation (thick coloured dashed line). The error bars at the borders of the windows indicate the standard deviation in the "start" and "end" times of the fountains (for more details see Table S1, Fig. S16). Attack types are schematically illustrated on top of (**D**). **E** Comparison of the collective prey recovery time $\tau$ between front, back, and side attacks (13 attacks by 10 marlins). The box spans from the first quartile to the third quartile, with a vertical line inside the box indicating the median value. A small red cross marks the mean value. The markers for individual events are consistent with event IDs as indicated in Supplementary Materials. Statistically significant difference among the attack directions is stated by the Kruskal-Wallis test followed by the Tukey test for multiple pairwise comparisons marked with $p < 0.05$ (*), ns = non-significant; ($p \approx 0.035$ for back-vs-side, $p \approx 0.033$ for front-vs-side, and $p = 0.9$ for front-vs-back attacks).

where animals attempt to flee while maintaining the predator inside their visual fields, inducing angles of escape that depend on the predator's position[10,11]. Similar to other escape rules of solitary prey[12–15], these models suggest prey can maximise their distances from the predator, but do not consider the role of social interactions during these escape manoeuvres. Indeed, optimal escape rules are likely to be affected by social interactions, given empirical studies show grouping prey often flee with more uniform escape trajectories than solitary ones[16]. No models, however, have considered how groups should optimise their escape trajectories to escape from the predator while interacting with others during an attack.

Fountain effects may also be the outcome of predators attempting to attack groups in ways that improve the likelihood of breaking groups apart or catching prey. Indeed, fountain effects are thought to occur when predators attack the centroid of groups from behind[17], as such attack directions may be a mechanism by which predators can break apart groups, improving capture success[18]. However, in the wild[4,19,20] as well as in the laboratory[21], predators often target groups from other angles (e.g., from the side or the front of groups). If the direction of an attack (in relation to a prey group's orientation) impacts the pattern of the escape responses produced[22] or the distance predators can approach individual prey, predators should target groups from particular directions. In response, prey should adopt robust

escape rules that maximise their survival chances across a range of attack scenarios.

Here, we investigate the mechanism and function behind the fountain effect in light of the conflicting interests of predators and prey. We do this by analysing aerial drone footage of striped marlin (*Kajikia audax*) attacking schools of sardines (*Sardinops sagax caerulea*) in the open ocean, quantifying fountain escape patterns and their dynamics following attacks on the groups from different directions. To understand the risks and rewards of particular escape rules and attack strategies, we extend a generic, spatially-explicit agent-based model of predator-prey interactions based on stochastic differential equations[23] to reproduce the fountain effect. Despite its common occurrence in nature, the fountain effect has rarely been captured in simulation studies[24–27], often involving hard-coded constituting elements or the absence of numerical analysis preventing an objective comparison with empirical data. Our model not only qualitatively captures properties of the fountain evasion, but also explains the advantages of using particular social escape rules. The escape rules used trade off maximising the distance of individual prey from the predator while minimising the time taken to rejoin the school after an attack. Moreover, the model makes predictions about why predators are observed attacking groups from particular directions. Overall, we highlight how fountain effects are the outcome of how predators

or prey are attempting to attack or avoid capture in ways that benefit their respective success, offering an explanation for the emergence of fountain effects during predator-prey conflicts.

## Results
### Empirical data analysis
We recorded aerial footage of wild predation events on a school of Pacific sardines by striped marlins, using unmanned aerial vehicles (DJI, Phantom 4 Pro V2.0) flying at 20 m altitude, 10-30 km offshore Baja California, Mexico. During 19 minutes of the video recording (at 30 fps), we observed $n = 136$ attacks launched by individual marlins (see Supplementary Note 1, Supplementary Fig. S1) from different directions with respect to the school's orientation, dashing one at a time through the prey school of ~ 100 individual sardines. In 104 out of these 136 attacks, we were able to estimate the directionality of the dash by measuring the attack angle $\alpha$ from the heading direction of the prey school's centre of geometry to the predator's head (Fig. 1D-top). We refer to the values of $\alpha$ closer to 0°, 90°, or 180° as attacks from the front, side, or back of the prey school, respectively. Attacks from the side of groups ($\alpha \in [90°, 100°]$) were the most common, followed by attacks from the back ($\alpha \in [150°, 180°]$), with attacks from the front of groups ($\alpha \in [0°, 20°]$) being the least common (Fig. 1B).

We qualitatively categorise escape manoeuvres by prey schools as "fountain" or "non-fountain" evasions (see Methods, Supplementary Figs. S2–S5). Fountains were defined as the prey splitting into two subgroups that moved in opposite directions to the predator, before subsequently rejoining behind the predator while turning to face its direction. Fountains occurred in the majority of attacks ($n = 67$ "fountains" versus $n = 37$ "non-fountain" evasions). Other non-fountain evasions involved the group sharply turning away without splitting, or the group being attacked from the front without splitting and turning away (see Supplementary Fig. S5).

Unlike in previous laboratory studies[11,21], we observed fountain effects predominantly when attacks came from the side of the school (Fig. 1C); the direction that predators were most likely to attack from (Fig. 1B). Non-fountain evasions were most likely to be occurring when the prey were attacked from behind (Fig. 1C); the second dominant attack direction (Fig. 1B). To understand why attacks from the side and from behind of prey groups may be the most dominant ones, we quantified the properties of fountain manoeuvres when groups were attacked from different directions.

### Self-organised dynamics during and post-attack
Using custom Matlab code, we extracted the escape angles of the prey and their relative positions to the passing predator before and during a fountain effect. We were only able to quantify fountain effects this way in $n = 30$ attack sequences, owing to the poor visibility of individual fish in some videos due to the sea conditions. To ensure compatibility with the computational model (see below), we restricted our quantitative analysis to instances where the predator was moving in a straight line through the school without sharp turns during the attack ($n = 14$). In all other cases, the high variability of the predator turning manoeuvres in terms of direction, magnitude and timing, together with a limited number of observations, prohibited a quantitative comparison with model simulations.

For each attack, we labelled the head and tail of 20 evenly distributed and randomly sampled sardines, as well as the head and dorsal fin of the attacking marlin (Fig. 1A), every third frame (videos filmed at 30 fps) of each sequence. The sequences contained several frames *before* and *after* the fountain effect occurred. We defined a "fountain" window $t_w$ based on the relative position of the predator in the prey school. We defined the "fountain" start, $t_{start}^{fnt}$, as 3 frames (0.1 s) before the predator's bill is inside the prey group, defined by the polygon over the fish that are located along the school's edges (Fig. 1A). We defined the "fountain" end, $t_{end}^{fnt}$, when the predator's head (mouth) exited the prey school.

To analyse the directional organisation of individuals in the "fountain" over time, for each attack sequence, we quantified the degree of collective order through polarisation $\Phi$ as the mean of the velocity orientations (see Methods) of sampled sardines in each frame, for each attack direction

separately (Fig. 1D). A significant decline in $\Phi$ indicates either fission or fusion events during and after the "fountain" window $t_w$, respectively. Since the amount of annotated frames differed between the attack sequences, to align different attacks in time, we perform a time shift such that the minimum value of $\Phi$ (Fig. 1D) occurred at the same time between attacks (for details see Supplementary Table S1). Note that despite 20 fish being sampled uniformly from the school because the school can split unequally (e.g., see Supplementary Fig. S6b), the average value of $\Phi$ can vary depending on how many individuals are on each of the split sides (on average, the fraction of annotations on split sides ranges in the proportion 35–65%).

The attacks from the back and from the front of the prey school were both described by the largest drop in polarisation compared to the attacks from the side. The largest decline in polarisation for the front attacks occurs on average after the predator leaves the school, highlighting that prey reverse the motion direction of the group after the predator has exited the school. That is, the prey school makes a 180-degree change in the orientation compared to the pre-attack. In the case of the back attacks, the decrease in polarisation is mainly observed during the "fountain" itself (i.e., when the predator is still inside the school), resulting from the prey splitting in opposite directions during the attack. The decline in $\Phi$ after the predator left was far less pronounced in the side attacks than in the front or back attacks. This is because for the side attacks, after the prey split, one of the group's divided sides aligns with the other side without the two sides having directly opposing orientations towards each other (see Supplementary Fig. S6).

To quantify potential advantages gained for both predators and prey from attacks coming from certain angles, we quantified how long it took schools to reconfigure following an attack. Shorter recovery times may be advantageous to prey by reducing the likelihood of being re-attacked during a manoeuvre. Longer collective prey recovery times, on the other hand, could be advantageous to the predator by enhancing the disruption and fragmentation of prey groups, undermining their collective defences. To quantify the time it takes the prey school to reorganise after an attack, we propose the metric of *recovery time* $\tau$ (see Methods), which indicates the amount of time needed for the prey after an attack (i.e., after $t_{end}^{fnt}$) to get back to its initial pre-attack state (i.e., before $t_{start}^{fnt}$). In the context of Fig. 1D, the latter corresponds to the highly aligned state, $\Phi \rightarrow 1$. We find that the recovery time from a side attack is less than the recovery time of other attack directions (Fig. 1E), with the recovery from the front and back attacks taking similar amounts of time. Altogether, therefore, attacks from the side are most common, but the group can return to an aligned state more quickly when attacks come from this direction. Attacks from the back and front, on the other hand, see the largest changes to the group's directional organisation and further take the group longer to return to an aligned state.

### Modelling the fountain effect
We use a spatially-explicit, stochastic agent-based model of predator avoidance adapted from[23]. For simplicity, we restrict ourselves to a two-dimensional set-up. This simplification is empirically justified as the prey schools in our field observation are confined close to the water's surface, and the prey evasion manoeuvres, and in particular the collective fountain responses, take place predominantly in a horizontal plane closely below the surface (see Fig. 1A). With our model, we explored what simple heuristics individual prey could use that ultimately lead to the collective dynamics observed in the empirical data above. In particular, we included a discrete escape manoeuvre into the avoidance force from the predator based on the concept of a *fleeing angle* $\Delta\alpha_{flee}$ (Fig. 2A). The response to the predator is modelled by a linear, distance-dependent repulsive force ("fleeing force"). For a vanishing fleeing angle $\Delta\alpha_{flee} = 0°$, the escape response occurs directly away from the predator along the vector $\overrightarrow{r}_{pi}$ connecting the position of the predator and the prey agent $i$. For a finite fleeing angle, the fleeing direction is rotated by $\Delta\alpha_{flee}$ in the direction away from the predator's velocity vector $\overrightarrow{v}_p$ (Fig. 2A).

While directly escaping a predator, the prey agents align with other neighbours, although the strength of the flee $\mu_{flee}$ is set as the strongest among other social forces (see Supplementary Table S2). Note, given that the

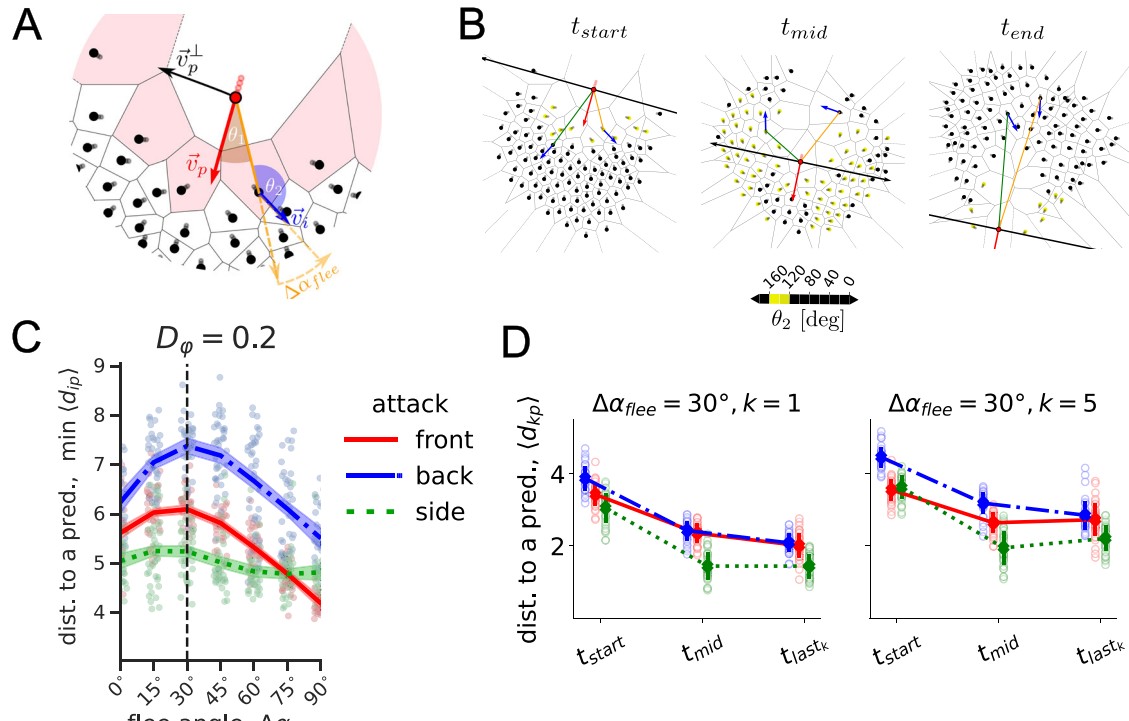

**Fig. 2 | Optimal flee angle. A** A snapshot from computational simulation modelling the "fountain effect". The predator (red point) interacts with its first shell of Voronoi neighbours (black dots in pink cells), which are referred to as *direct responders*. All direct responders flee away with a repulsion force along the respective radius-vector $\vec{r}_{pi}$ rotated on $\Delta\alpha_{flee}$ in the direction away from the predator's orientation $\vec{v}_p$. **B** The state of the simulation at the "start", "middle", and "end" of the "fountain". These time points are defined by the relative position of the prey agents (black dots) to the black line, described by the vector $\vec{v}_p^{\perp}$. The predator moves with velocity $\vec{v}_p$ aiming at the centre of mass of the prey school, which is reached by $t_{mid}$. While flee angle as an "internal" behavioural parameter $\Delta\alpha_{flee}$ is set to 30°, due to the impact of social interactions the resulting orientation angle $\theta_2$ does not equal 150° for all individuals but occurs to be in the range $\theta_2 \in [120°, 160°]$ for the prey agents coloured in shades of yellow. **C** The minimal distance to the predator $\min\langle d_{ip}\rangle$ (averaged across all prey individuals $i$), which was ever achieved during the "fountain" window $[t_{start}, t_{end}]$, depending on the flee angle $\Delta\alpha_{flee}$ with orientational noise intensity $D_\varphi = 0.2$ for simulated front, back, and side attacks. The curves are created by local regression with shading areas of 95 % confidence interval based on $n = 40$ independent simulation realisations for each attack type. The dashed vertical line shows the theoretical optimal flee angle $\Delta\alpha_{flee}^*$ of a single escaping fish, as by Hall et al.[11]. **D** Time-dependent distance changes between the predator and $k$ closest prey. Distance to the predator $\langle d_{p,k}\rangle$ averaged across its $k$ nearest neighbours, using $\Delta\alpha_{flee} = 30°$, at the start, middle, and end of the "fountain". The end is specified by the time $t_{last_k}$ when there are only $k$ prey agents left in front of the predator. The error bars indicate the standard error over $n = 40$ independent simulation realisations.

model also considers fluctuations in the heading of the prey with the intensity $D_\varphi$, the flee angle $\Delta\alpha_{flee}$ does not necessarily complement the resulting orientation angle $\theta_2$ of the prey. Namely, although we set the flee angle in the model, it defines the direction of the flee force, but not the final angle at which the agent flees. Since the fleeing force is additive to other social forces and orientational noise, despite being the strongest force among others, the final combination of forces defining the agent's orientation $\theta_2$ is not necessarily $\theta_2 = 180° - \Delta\alpha_{flee}$ (see Fig. 2B with $\Delta\alpha_{flee} = 30°$). To ensure prey fusion after the split, we also replaced a distance regulating social force from[23] with a simple spring-like linear attraction-repulsion function (see Methods). As a result, we expanded the functionality of the original model[23], where the fountain effect had not been a priori observed. The model operates on the assumption that social forces modify only the prey's orientation, keeping the speeds of both predator $v_p$ and prey $v_0$ constant ($v_p = 2v_0$). The model speeds choice is consistent with the subsampled estimation of the predator-prey speed ratio ($v_p$ : $v_0$) in the empirical data (mean: 1.85, median: 1.88, range: [1.51, 2.14], standard deviation: 0.21) and does not show significant variations in the simulation results for the range $v_p = [1.5v_0, 2.0v_0]$ (see Supplementary Note 2, Supplementary Fig. S7). Independent of the attack angle, we consider that the predator is moving linearly towards the updated position of the prey's school centroid and switches to moving straight with its current velocity $\vec{v}_p$ when it enter the $\epsilon$-neighbourhood of $\vec{r}_{com}$ ($\epsilon := \vec{r}_{com} - \vec{r}_p \leq 1$). With this approach, we aim to relax the impact that variations in the predator trajectory have on the collective response of the prey, to comprehend its self-organised dynamics and to compare with the empirical data.

We consider social interactions between prey which are maintained during predator avoidance. In particular, focal individuals directly interact with the first shell of their Voronoi neighbours, whether these are other prey agents or the predator. Voronoi networks have been shown to approximate visual interactions in fish schools[28] and therefore provide a biologically realistic interaction neighbourhood. In the following, we refer to the first Voronoi shell of the predator's neighbours as *direct responders* (see Fig. 2A). That is, within the fleeing range $R_{flee}$ these agents are affected by the flee force. However, the "fountain effect" is primarily observed when the flee strength $\mu_{flee}$ is sufficiently strong (see Supplementary Note 3, Supplementary Figs. S8, S9), such that social forces become negligible for directly responding individuals, similar to previous models, e.g.,[11,27].

To allow comparison with the empirical data, we defined a "fountain" window $t_w$ in the model based on the relative position of the prey to the orthogonal predator's velocity vector $\vec{v}_p^{\perp}$. Namely, the vector $\vec{v}_p^{\perp}$ divides the space into the regions 'ahead' of and 'behind' the predator (Fig. 2B). We define the "fountain" has started, $t_{start}^{fnt}$, as soon as there is at least one prey individual 'behind' the predator while the rest of the group is 'ahead' of it. When the predator exits the school, we define the "fountain" end, $t_{end}^{fnt}$, such that there is only one prey individual 'ahead', separated from the rest of the group which is left 'behind' (see Methods for details).

**Optimal flee angles and optimal angles of attack.** We first tested the prediction that there is an optimal flee angle of the prey that maximises the distance of prey fish from the predator as it attacks[11], while socially interacting with others. To do this, we performed systematic numerical

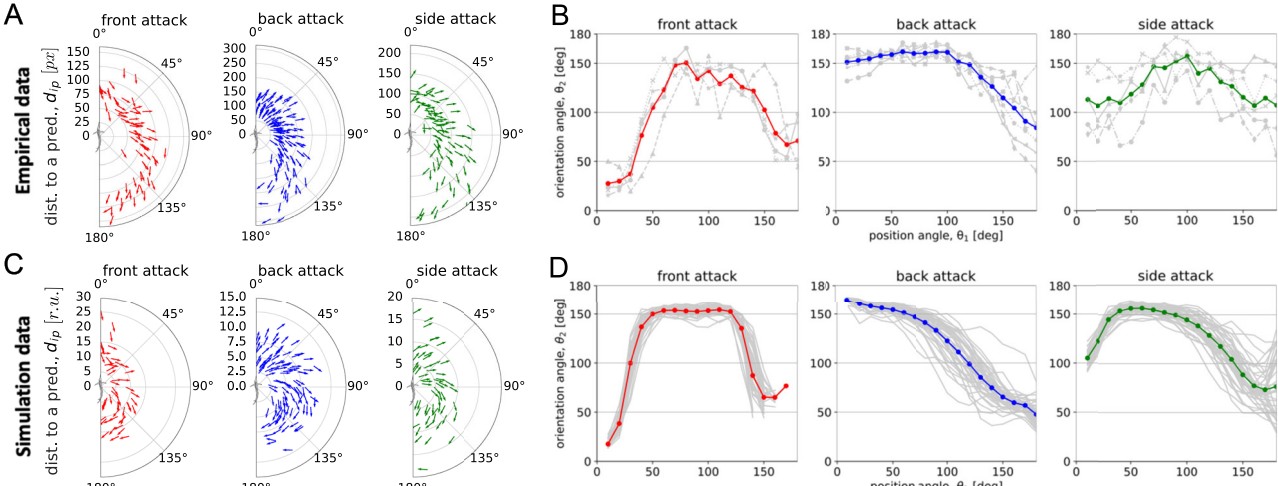

**Fig. 3 | Self-organised dynamics of the fountain manoeuvre.** Polar plots illustrating aggregation of the attacks, where for each single attack the mean prey position and mean prey orientation in the increment of $\Delta\theta_1 = 10°$ away from the predator's head (placed in the origin) are computed over $[t_{start}^{fnt}, t_{end}^{fnt}]$. The amount of arrows in each sector $\Delta\theta_1$ is defined by the amount of respective attacks in (**A**) empirical and (**C**) simulation data with $\Delta\alpha_{flee}^* = 30°$. Radial y-axis indicates relative distance obtained from video data in pixel adjusted to the dimension of the manoeuvre.

Relationship between prey's position angle relative to the predator ($\theta_1$) and prey's orientation (swimming) angle ($\theta_2$) in (**B**) empirical and (**D**) simulation data with $\Delta\alpha_{flee}^* = 30°$. Each grey line represents a single fountain instance, while the thick line in colour shows the average over the instances. The results in (**D**) are based on 40 implementations, while in (**C**), to prevent visual bias compared to the empirical data, we sampled an equivalent number of attacks as in (**A**).

simulations of our model across different flee angles $\Delta\alpha_{flee}$ (ranging from 0° to 90° with the step of 15°) for varying noise levels $D_\varphi$ and different approach angles of the predator denoted as *front*, *back*, and *side* attacks, including their intermediate variations (see Supplementary Note 4). We evaluated the optimality of the flee angle in terms of the mean distance of prey from the predator within the time window of the fountain effect. That is, the mean distance $\langle d_{ip}$ of all prey $i = 1, \ldots, N$ to the predator is computed at each time step of $t_w$, and the lowest mean distance of prey to the predator over the attack, noted as $\min\langle d_{ip}\rangle$, is selected. In this way, the minimal mean distance is strongly correlated with the maximal risk of predation within the school[18,29,30].

The model shows that regardless of the attack angle, the minimum distance from a predator peaks at around the same flee angle $\Delta\alpha_{flee}^* = 30°$ (Fig. 2C). With this optimal flee angle, predators are more likely to get closer to prey when attacking from the side of groups, compared to the front and back of groups. This pattern is largely consistent regardless of the noise intensity of the flee angle (see Supplementary Note 4, Supplementary Fig. S10) and is consistent with previous theoretical results[10,15,31–33] for a single deterministic prey agent, i.e., in the absence of noise and interactions with the conspecifics. However, we note that here, the optimal flee angle $\Delta\alpha_{flee}^* = 30°$ emerges in the absence of an explicit blind angle in prey as it was suggested based on theoretical consideration in Hall et al.[11], but within the context of the spatial self-organisation of the Voronoi interaction network. With an additional blind angle on top of the Voronoi neighbourhood, the optimal flee angle remains robust relative to the shift imposed by the blind angle in the range of [15°, 30°], i.e., the resulting flee angle is the sum of the blind angle and the $\Delta\alpha_{flee}^*$ (see Supplementary Note 4, Supplementary Fig. S10).

We also evaluated the mean distance to the predator $\langle d_{p,k}\rangle$ of only the $k$ nearest neighbours at the start, middle, and end of the "fountain" (Fig. 2D). This allows us to assess at which moment of the fountain the predator gets closest to individual prey, and therefore the moment of greatest risk to individual prey. For the closest prey, i.e. when $k = 1$, the predator generally gets closest to the prey over $[t_{start}, t_{last_1}]$ regardless of the approach angle $\alpha$. As with the group metrics, with an optimal flee angle $\Delta\alpha_{flee}^* = 30°$, the back attack is more advantageous for the prey among other attack angles at the start $t_{start}$, but the distance to prey levels off, so that by the end $t_{last_1}$ of the "fountain", this distance is similar to that of front attacks. Side attacks, on the other hand, result in the lowest $\langle d_{p,1}\rangle$ at all times compared to others, with no additional decline in $[t_{mid}, t_{last_1}]$.

For the $k = 5$ closest individuals to the predator (i.e., the average number of direct responders to a predator at a time), the distance $\langle d_{kp}\rangle$ from the predator is initially larger for back attacks than side and front attacks at $t_{start}$ and $t_{mid}$. However, owing to an increase in $\langle d_{p,5}\rangle$ towards the end of the fountain for front attacks, this distance ends up being equivalent to distances in back attacks by the end of the attack. Overall, side attacks generally result in the lowest distance to the predator for the closest $k = 5$ individuals and are consistent with the ones in Fig. 2C, indicating that the average minimal distance to the predator $\langle r_{ip}\rangle$ over the group is correlated with the distance of a single ($k = 1$) individual. Our simulations, therefore, predict that predators will try to attack schools from the side as it allows them to get closer to prey *during* and *at the end* of the attack, during which the school performs the fountain manoeuvre.

**Impact of the flee angle on self-organised dynamics.** To test whether our model incorporating an escape angle of $\Delta\alpha_{flee} = 30°$ captured the dynamics of the fountain effect observed in the empirical data, we computed the respective mean orientation angle of the prey fish ($\theta_2$) in relation to the fish's position relative to the predator's head ($\theta_1$) at each 10° in both the empirical data and model (Fig. 3). Figure 3A shows the observed evasion reactions over the duration of the "fountain effect" plotted relative to the predator (the origin of the coordinate system) for all analysed encounters and each attack direction. For each attack encounter, 20 random fish were annotated each third frame up until the predator left the school. Each arrow in Fig. 3A represents the average position and orientation of the fish in each encounter at 10° increments of $\theta_1$ relative to the predator. The amount of arrows in each 10° angular sector is defined by the amount of respective attacks. This way, Fig. 3A illustrates the aggregation of all attacks of each type, highlighting the similarity of the encounters in their response. To ensure a fair visual perception of the illustrated empirical and simulation data (Fig. 3A vs. Figure 3C), we subsampled the simulation data to match the amount of empirical data. That is, in Fig. 3C 20 prey-agents were uniformly selected at random from the prey group at each iteration over "fountain" duration and the same number of attacks for each type as in the empirical data was used. Overall, Figs. 3A, C describe the average escape angles over $t_w := [t_{start}^{fnt}, t_{end}^{fnt}]$ as the predator passes through and exits the school.

Figure 3B shows that, in case of the back attack, for $\theta_1 \in [0°, 100°]$ there is a consistency in the prey escape angle around $\theta_2 \approx 150°$. As the prey get

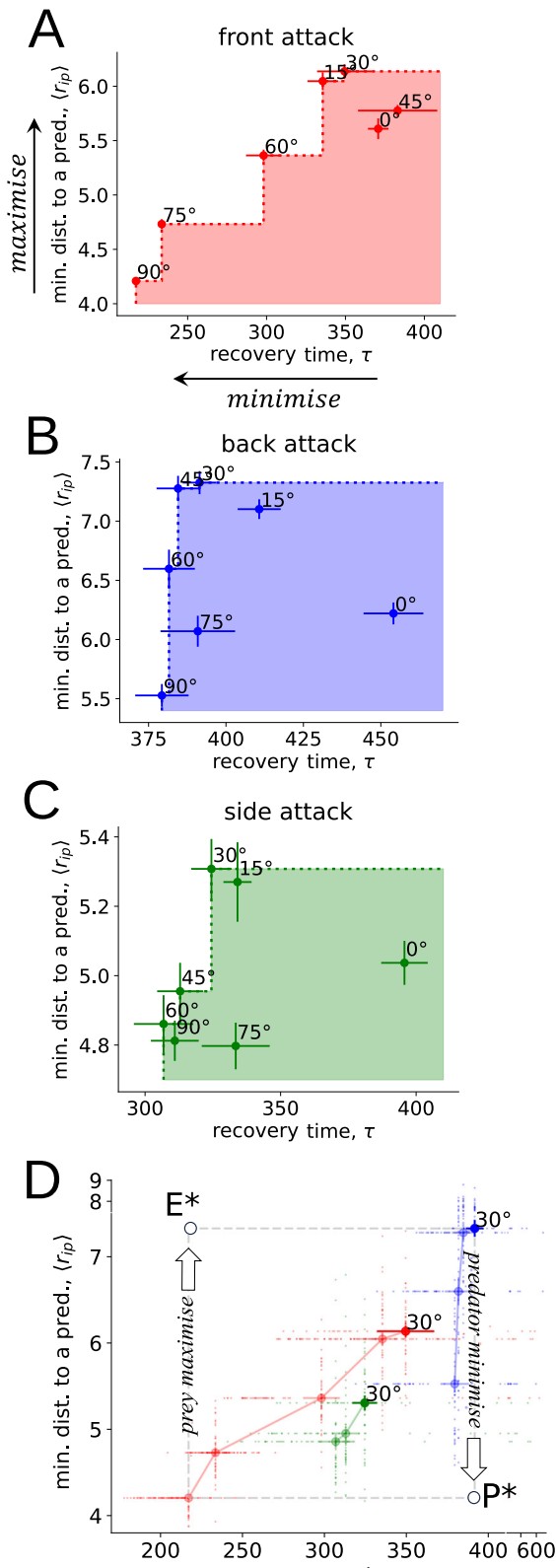

**Fig. 4 | Conflict between collective recovery time and distance from the threat.**
**A–C** Pareto charts of flee angles $\Delta\alpha_{flee}$ as prey movement strategies, with the utopia point as the maximum value of minimal averaged prey distance to a predator $\langle r_{i,p}\rangle$ and the minimum value of collective recovery time $\tau$ (indicated by arrows in A). Each dot represents the mean value, and the horizontal and vertical bars indicate the standard errors for the corresponding metric over $n = 40$ independent simulation implementations. Non-dominated solutions that form the Pareto front and reveal the trade-off lie on the chart's "steps" covering the others (in shaded areas).
**D** Perspective on the Pareto fronts of flee angles, conditioned on the prey escape, as predator's attack strategies referred to front (in red), back (in blue), and side attacks (in green). P* denotes the ideal solution for the predator, and E* for the prey.

initial relative position at the start of the attack. Front attacks are characterised by the sharp increase in $\theta_2$ from approx. 25° up to 150° for the individuals in front of the predator $\theta_1 \in [0°, 75°]$. The escape angle levels up at around 150° for $\theta_1 \in [75°, 125°]$ as the predator gets inside the school and declines for $\theta_1 > 125°$ as the prey reunite behind its tail. Side attacks are described by greater variability across the encounters, compared to back and front attacks. Similarly to front attacks, for $\theta_1 \in [75°, 125°]$ the average observed prey escape angles in the case of side attacks vary around $\theta_2 \approx 150°$. Figure 3C, D demonstrate that incorporation of $\Delta\alpha_{flee} = 30°$ into the model qualitatively reproduced collective patterns of escape in the simulation as observed in the empirical data (Figure. 3A, B).

**Optimal post-attack recovery time.** Although escaping from a predator at $\Delta\alpha_{flee} = 30°$ increases the distance between the prey and the predator (Fig. 2C), such a relatively small flee angle can also result in longer arched trajectories[27]. This, in turn, can lead to the two split subgroups taking longer time to recover (i.e., recoalesce) after being split (Supplementary Fig. S11). Shorter recovery times $\tau$ are likely to be beneficial to prey, as this allows them to exploit the benefits of larger groups, such as the dilution and confusion effects, sooner. Consequently, achieving shorter recovery times while retaining larger distances from the predator may be incompatible objectives for the prey. To assess how each flee angle $\Delta\alpha_{flee}$ performs across these two objectives, we conducted a multi-objective analysis of our simulation results for different attack angles (Fig. 4A–C). For each attack direction, we find a set of fleeing angles $\Delta\alpha_{flee}$ that creates a trade-off between the distance to the predator and the recovery time, forming the so-called Pareto-optimal solutions (see Supplementary Note 5). There, improvements in one metric, i.e., increasing the distance from the predator, can lead to the worsening in the other, i.e., prolonging the recovery time. If we assume that there is no preference for one of the two aforementioned objectives (i.e., prey weight each objective with the same importance), the flee angles $\Delta\alpha_{flee}$ that ideally balance recovery time $\tau$ with distance$\langle r_{i,p}\rangle$to the predator for each attack direction (front, back, side) are $\Delta\alpha_{flee} \in \{60°, 45°, 45°\}$, respectively (see Supplementary Note 5, Supplementary Table S3). However, the distance between prey and predator during an attack is a more immediate threat to life for the prey, and therefore this objective is likely to take stronger importance over decreased recovery time. If there is a strong preference for maintaining larger distances from the predator $\langle r_{i,p}\rangle$ rather than on decreasing recovery times, we find that the respective solution on all Pareto fronts corresponds to $\Delta\alpha_{flee} = 30°$, regardless of the attack angle (Fig. 4D). In this way, $\Delta\alpha_{flee}^* = 30°$ can be considered as an efficient heuristic, which prioritises maximisation of the distance from the predator. In other words, alternative Pareto-optimal solutions, different from $\Delta\alpha_{flee} = 30°$, enhance recovery time but tend to reduce the distance from the predator.

We can further analyse the Pareto-fronts of prey escape behaviour from the perspective of the predator's attack strategies, referred to as front, back, and side attacks (Fig. 4D). Conditioned on the Pareto-optimality of the prey's response, we find that the flee angle $\Delta\alpha_{flee} = 30°$ creates a conflict in the effectiveness of the predator's attack strategies and prey avoidance. Indeed, predators are the least effective at attacking prey groups from behind, as prey can get furthest away from the predator when they are attacked from this direction, despite a longer recovery time (increasing $\tau$). Our model instead

into the zone behind the predator $\theta_1 > 100°$, there is a sharp decline in $\theta_2$ indicating that the fish are turning to reunite with the other half of the school formed during the split. These results are consistent with previous work by Hall et al.[11] performed in the laboratory conditions. Since the fish orientation and positions are computed relative to the predator, the results vary depending on the attack direction due to the predator's movement and its

predicts that predators would be most effective if they attacked groups from the side (Fig. 4D), despite the fact that this attack direction results in the shortest recovery time $\tau$ for prey compared to other attack directions (Fig. 4D; consistent with our empirical evidence, Fig. 1E). While shorter recovery times are not favourable for the predator, side attacks remain closest to the ideal solution for the predator across bi-objective space (Fig. 4D; see also Supplementary Note 5 for more details, Supplementary Fig. S12). Furthermore, although our results suggest back and front attacks are suboptimal for the predator with respect to the distance to the prey, predators receive a secondary benefit from front or back attacks because of the related increase in the prey's recovery time, potentially leading to increased dispersion.

## Discussion

Using an empirically motivated agent-based model of predator-prey interactions, we show that by combining social interactions with a fleeing angle, groups of prey can produce "fountain effects", with prey avoiding predators by splitting into two subgroups before rejoining together. By viewing prey escape behaviour from a multi-objective perspective, we find a range of flee angles that create a trade-off between the distance that prey can keep from the threat and the time needed to re-organise back into a cohesive, polarised group following an attack. A flee angle of 30°, however, represents a robust and optimal choice across a range of attack directions to maximise the prey's distance from the predator, albeit resulting in longer recovery times for the group. These results highlight how attack and avoidance strategies give rise to collective escape patterns in contexts where prey have no place to hide and when predators aim to break up prey groups to disperse and isolate individuals[34,35].

Predators often attempt to fragment prey schools, as this reduces school size, which can increase capture success[18,29,36]. Our empirical data collected under natural conditions show that predators generally attacked schools from the side and from behind, with side attacks most frequently leading to "fountains". Previous studies using artificial predators have associated fountain effects with attacks from the back of the schools[11]. While fountains from this attack direction can occur, "non-fountain" evasions, where the entire school evaded the attack cohesively in one direction rather than splitting and re-joining (see Supplementary Fig. S5), predominantly occurred when groups were attacked from the back (Fig. 1C). As our empirical data show, prey attacked from the back are able to adjust their escape to avoid fragmentation altogether, preventing the predator from moving through the school. According to our modelling predictions, despite side attacks resulting in shorter recovery times for prey (as also observed empirically), they represent a better solution in the predator's bi-objective space compared to front or back attacks. Attacking from the side not only enables the predator to get closest to the prey school, which is likely to enhance its capture success (e.g.,[18,29]), but also represents the best compromise between approaching the prey while increasing their recovery time. The latter can be viewed as a capitalisation of the prey's response (i.e., producing a fountain effect) when predators attack from the side.

From the prey's perspective, prey should prioritise maximising their distance from the predator over minimising recovery time, as being close to a predator represents an immediate threat to life. Under these assumptions, a flee angle of 30° was predicted to be the best escape heuristic (Fig. 4A–C). Critically, this 30° flee angle emerged in the absence of an explicit blind angle of the prey and maximised distance to the predator across all attack directions. Therefore, this simple heuristic appears to maximise individual survival success across a range of attack scenarios. On the other hand, if we assume that prey seek a trade-off between distance to a predator and recovery time, our model predicts the best compromise would be to flee away at an angle of 45° (Supplementary Note 5, Supplementary Table S3). This choice would be robust for scenarios when prey are attacked from behind and from the side, but not from the front of the school. Prey may also prioritise distance by maximising the time to collision between themselves and predators. Some studies in human crowds, e.g.,[37], suggest that time-to-collision may play an important role in the social interaction, which is

equivalent to relative speed dependent repulsion[38], and that these type of interactions may play a role in predator evasion. However, these interactions have so far not been considered with a fleeing angle, and it is unclear under what conditions, or whether at all, they can produce a robust "fountain effect". This should be addressed by future research.

Our results suggest there is an ongoing arms race between predators and prey in the strategies they are using to improve their own success at the expense of the other. If prey use an optimal flee angle of 30°, predators are more effective in attacking from the side of the schools, while prey are more effective in escaping when attacked from behind. While prey are relatively more manoeuvrable than predators[8], and therefore should have more control over which direction they are attacked from, our empirical data show that attacks from the side and behind were the two most common attack directions (Fig. 1B), potentially highlighting a battle for the attack directions each party prefers. While attacking from the back or side benefits prey or predators respectively, these outcomes introduce indirect conflicting consequences arising from the existing trade-offs. In particular, back attacks provide an indirect, rather than the primary, advantage to the predator in increasing the recovery time of the prey if a fountain is produced (despite prey keeping larger distances from the predator), while side attacks allow groups to recover faster (despite predators getting closer to the prey).

In general, while significant research effort has gone into understanding arms-races between predators and prey from an evolutionary perspective, little effort has been made into understanding the role of complex spatio-temporal dynamics of collective prey responding to (collective) predators. Recently developed technological capabilities (combining drone technology and video tracking) allow researchers to obtain quantitative data on complex predator-prey interactions at high spatial and temporal resolutions[39]. Nevertheless, interpreting these data remains a fundamental challenge. Our modelling approach provides a means to address the inherent difficulty of inferring specific behavioural rules like the "desired" fleeing angle $\Delta\alpha_{flee}$ purely from trajectory data. In these systems, the observed movement of individuals is a product of individual movement decisions, social interactions and stochastic effects. Thus, any observed fleeing angle that can be extracted from individual escape trajectories may significantly deviate from any true internal $\Delta\alpha_{flee}$ rule. Here, modelling approaches combining evolutionary dynamics with spatially-explicit multi-agent dynamics are promising. They allow for a complementary investigation of the intricate interplay of spatio-temporal self-organisation and function, and on the emergence of specific spatio-temporal collective response patterns as evolutionary stable strategies (ESS)[23]. According to[5,23], the combination of high alignment strength with a strong flee strength (like in our model) implies that the "fountain effect" can correspond to an ESS. While further work is needed to confirm this, it is plausible to hypothesise its validity based on the ecological context. Nevertheless, an understanding of the rules underlying these collective escape patterns, and in particular the interplay between prey avoidance rules and predator attack angles, is relevant to a variety of predators that attack grouping prey. Furthermore, this approach will inform research about how groups of humans or autonomous vehicles can avoid threat and improve coordination[27,40,41].

Overall, our modelling predictions suggest that empirically prevalent attack directions can be to the advantage of either predator or prey. This indicates an ongoing arms race, where both predators and prey continuously adapt their tactics to enhance their own success relative to the other. As predators demonstrate effectiveness in attacking the groups from the side, whereas prey show agility and effectiveness in evading attacks from behind.

## Methods
### Empirical data collection
Aerial video of striped marlin and sardine school predator-prey interactions was collected opportunistically from unmanned aerial vehicles (DJI, Phantom 4 Pro V2.0) flying at 20 m altitude, 10–30 km off the coast of Baja California, Mexico (N 24° 54.52–48.5', W 112° 34.46–23.52') from the 10.11.2021 to the 13.11.2021 during daylight hours. The recordings were in

3840 × 2160 pixel resolution using a WalimexPro ® circular polarization filter to enhance clarity. We complied with all relevant ethical regulations for animal use. The empirical data was purely observational and collected under approval from the Comisión Nacional de Acuacultura y Pesca (CON-APESCA, Mexico) permit PPF/DGOPA-024/20 and the Secretaría de Medio Ambiente y Recursos Naturales (SEMARNAT, Mexico) permits SGPA/DGVS/02460/18, SGPA/DGVS/01643/19, SGPA/DGVS/07490/20 and SGPA/DGVS/08074/21. The recommendations for fieldwork outlined by ASAB[42] were followed.

### Categorising "fountain" and "non-fountain" escape manoeuvres
Qualitatively, we relate "non-fountain" cases (see Supplementary Fig. S5) to the attack events when the prey school is performing a sharp turn to the predator's tail, but the whole school sticks together and does not split into two subgroups by either side of the predator unlike in the case of the "fountain effect" (see Supplementary Figs. S2–S4). We also quantified the shape of the prey group based on the points of the annotated polygon over the fish positioned on the edges of the school as the ratio of its convex hull perimeter to the polygon's perimeter to define the level of the group's convexity $C$ during the attack (Supplementary Fig. S13). Supplementary Fig. S14 shows the convexity measures of the prey group's shapes for "fountain" and "non-fountain" evasions in the middle ($C_m$) and at the end ($C_m$) of an attack. "Fountain" evasions are characterised by lower measures of convexity compared to "non-fountain" ones at the end of attacks, i.e., $C_e < 0.9$, which indicates a split in the group, confirming our qualitative classification. However, there are some borderline cases (typical for the front attacks) in the case of the "fountain" evasions, where the annotated group's shapes do not undergo significant convexity changes, as prey are splitting in a more streamlined manner around the predator (see for instance, Supplementary Figs. S3d, S4a). However, these are distinguishable through observation as the individual prey (more than 2 individual fish) are evidently located by both sides of the predator (Supplementary Fig. S3c) compared to the "non-fountain" cases (Supplementary Fig. S5f).

### Estimation of relative predator-prey speeds
An estimation of predator and prey speed was calculated by tracking the positions of both predator and prey using aerial video of attacks ($N = 13$). These attacks were selected based on two criteria: (i) the drone was stationary (corroborated with the flight log) and, (ii) the position of the predator and an individual sardine were visible at each consecutive frame during a full second of the attack (i.e., 30 frames or 60 frames depending on the video). The frames were exported from each video using VirtualDub[43] and imported into ImageJ[44]. A stereo camera (90 cm. in length), visible in each of the exported frames, was used as a scale object. Using the manual tracking plugin MTrackJ[45], the trajectory of the predator was measured by tracking the space between its pectoral fins at each frame. The trajectory of the sardine was measured by tracking the mid-point of its body for the same set of frames. The cumulative distance travelled by both predator and prey over every frame was calculated to get speed (cm/s): predator (mean ± se = 334.76 ± 11.74, range = [257.95, 397.75]); prey (mean ± se = 183.33 ± 8.05, range = [125.36, 228.62]). Predator speed was divided by prey speed to calculate the predator:prey speed ratio (mean ±se = 1.85 ± 0.06, range = [1.51, 2.14]).

### Agent-based model description
The prey agents are modelled as active Brownian agents $i = 1, \ldots, N$ with constant speed $v_i = v_0$ and an angular noise $D_\varphi$ based on the following stochastic differential equations of motion[23]:

$$\frac{d \vec{r}_i(t)}{dt} = \vec{v}_i(t) = v_0(\cos(\varphi_i), \sin(\varphi_i)) \tag{1a}$$

$$\frac{d \vec{\varphi}_i(t)}{dt} = \frac{1}{v_0}\left(F_{i,\varphi}(t) + \sqrt{2D_\varphi}\xi(t)\right), \tag{1b}$$

where $\vec{r}_i$ is the position vector of an agent $i$, $F_{i,\varphi}(t) = \vec{F}_i(t) \cdot \vec{e}_\varphi$ is a social force which is projected on the turning direction $\vec{e}_\varphi$ of an agent $i$ and $\xi(t)$ is a Gaussian white noise. The social force is composed as follows: $\vec{F}_i = \vec{F}_{i,alg} + \vec{F}_{i,d} + \vec{F}_{i,flee}$. The agents align to each other with $\vec{F}_{i,alg}$ to keep the same direction of motion, they repel from each other with $\vec{F}_{i,d}$ at short ranges (if $|\vec{r}_j - \vec{r}_i| < l$), and attract to each other at long ranges (if $|\vec{r}_j - \vec{r}_i| > l$) to stay cohesive:

$$\vec{F}_{i,alg} = \frac{1}{|\mathbb{N}_i|}\sum_{j \in \mathbb{N}_i} \mu_{alg} \cdot \left(\vec{v}_j - \vec{v}_i\right), \tag{2a}$$

$$\vec{F}_{i,d} = \frac{1}{|\mathbb{N}_i|}\sum_{j \in \mathbb{N}_i} k \cdot \left(\left(\vec{r}_j - \vec{r}_i\right) - l\vec{u}_{ji}\right), \tag{2b}$$

where $\vec{u}_{ji} = \frac{\vec{r}_j - \vec{r}_i}{|\vec{r}_j - \vec{r}_i|}$. The neighbourhood subset of an agent $i$ is denoted by $\mathbb{N}_i$ and defined by the first Voronoi shell of neighbours. If a predator $p$ is a neighbour of an agent $i$, $p \in \mathbb{N}_i$, and $r_{pi} < R_{flee}$ ($r_{pi} := |\vec{r}_{pi}| = |\vec{r}_p - \vec{r}_i|$) the agent $i$ flees away from a predator according to

$$\vec{F}_{i,flee} = -\mu_{flee} \cdot \hat{r}_{flee} \tag{3}$$

with $\hat{r}_{flee}$ being the fleeing direction unit vector, which is given by the direction directly away from the predator $\hat{r}_{pi} = \vec{r}_{pi}/r_{pi}$ rotated by the additional fleeing angle $\Delta\alpha_{flee}$ towards the rear of the predator:

$$\hat{r}_{flee} = \mathbf{R}(\pm\Delta\alpha_{flee})\hat{r}_{pi} \tag{4}$$

with $\mathbf{R}(\alpha)$ being the rotation matrix by an angle $\alpha$ in 2D. The turning direction, i.e., the sign of the angle $\pm\Delta\alpha_{flee}$, is always directed towards the rear of the predator, corresponding to the opposite direction to the predators velocity $\vec{v}_p$. The predator moves with a fixed speed $|\vec{v}_p| = v_p = 2v_0$.

The Euler-Maruyama method[46] is used to simulate the presented above stochastic differential equations with the time-step $dt$.

The predator is initialised relative to the prey's group centre of mass $\vec{r}_{com}$ randomly within one of the defined angular regions at the distance of $R_p$ from the maximum remote prey agent from $\vec{r}_{com}$ in this region. This initial relative positioning defines the predator's *attack angle* (i.e., 0° ± 30° as front attack, 90° ± 30° as side attack, 180° ± 30° as back attack). The right-hand or the left-hand positioning of the predator to the prey group is assigned randomly. The predator moves linearly to the updated position of the prey's group centroid $\vec{r}_{com}$ at each simulation step, without noise in its velocity orientation $\vec{v}_p$. Once predator gets into the vicinity of the preys' centroid, i.e., $\epsilon := \vec{r}_{com} - \vec{r}_p \leq 1$, it continues moving straight at its current speed resulting in a straight-line trajectory after passing $\vec{r}_{com}$. Before the predator is initialised, the prey group evolves for $T_{eq}$ time units.

### Quantification of collective dynamics
For the empirical data, to estimate the degree of the collective order of the fish school before, during and after the escape, we computed polarisation $\Phi$ as the absolute value of the mean of the fish heading directions $\Phi(t) = |\sum_i^s \vec{u}_i(t)/n_s|$, where $n_s$ is the number of annotated fish in the frame $t$ and $\vec{u}_i := \vec{v}_{i,flee}/\sqrt{\vec{v}_{i,flee} \cdot \vec{v}_{i,flee}}$ with $\vec{v}_{i,flee} = \vec{x}_{i,head} - \vec{x}_{i,tail}$, where $\vec{x}_i$ is the annotated location of the fish $i$ head's/tail's coordinates (in pixels) in the frame.

**The start and the end of the fountain manoeuvre.** Before the predator launches the attack, all prey agents $i = 1 . . N$ are positioned in front of the

predator: $\overrightarrow{r}_{pi} \cdot \overrightarrow{v}_p < 0$. We defined the start of the "fountain effect" ($t_{start}^{fnt}$) as soon as at least one prey individual $i$ passes from frontal to rear relative position with respect to the predator, i.e., $\overrightarrow{r}_{pi} \cdot \overrightarrow{v}_p > 0$. Accordingly, we defined the end of the "fountain effect" ($t_{end}^{fnt}$) as soon as $\overrightarrow{r}_{pi} \cdot \overrightarrow{v}_p > 0$ is true for all $i = 1..N$. These criteria also apply to empirical data, where $t_{start}^{fnt}$ is defined by checking whether the annotated predator's bill tip is within a polygon defining the borders of the prey school (see Fig. 1A) using the Python module *mplPath*.

**Collective recovery time.** We computed the time-averaged mean polarisation $\overline{\Phi} = \langle\langle\Phi(t,s)\rangle_s\rangle_t$ of the prey group over $n_s = 40$ simulations conducted in the absence of disturbance (predator) as a baseline order state of the simulated school (see Supplementary Fig. S15a). The respective standard deviation $\overline{\sigma}$ was computed over the concatenated array containing all polarisation values $\Phi(t,s)$ of the undisturbed prey at time $t$ and each simulation $s$. In the simulations with a predator, we categorised the values of $\Phi(t)$ below the lower control limits, denoted as $LCL_\Phi := \overline{\Phi} - 3 * \overline{\sigma}$, as indicative of a *perturbed* state of the prey group, and $\Phi(t) \geq LCL_\Phi$ as of a *recovered* state of the prey (for details see Supplementary Note 6, Supplementary Fig. S15). To enhance the temporal stability of the data and minimise the impact of fluctuations, we applied the Savitzky-Golay filter on each time step of the time series difference $\hat{y}(t) = LCL_\Phi - \Phi(t)$. As a result, based on the smoothed signal $\hat{y}(t)$, the prey recovery time $\tau$ is the time point when $\hat{y}(t) > 0$ transitions to $\hat{y}(\tau) = 0$.

For the empirical data, the lower control limits $LCL_\Phi$ are based on the mean $\overline{\Phi}$ and $\overline{\sigma}$ computed over the polarisation values $\Phi(t < t_{start}^{fnt})$ before the start of the fountain manoeuvre, satisfying $\Phi(t,s) > 0.8$ to eliminate that the school can be still recovering from the previous attack and be in a perturbed state. The recovery time $\tau$ is set as a time point when polarisation values after the attack $\Phi(t > t_{end}^{fnt})$ satisfy $\Phi(t) \geq LCL_\Phi$ such that the standard deviation of the polarisation sequence $\Phi(t \geq \tau)$ is close enough ($\epsilon < 0.01$) to the standard deviation $\overline{\sigma}$ of the polarisation sequence $\Phi(t < t_{start}^{fnt})$ before the attack.

**Statistics and reproducibility**

Statistical analyses (Kruskal-Wallis test, Tukey's test for multiple pairwise comparisons, repeated measures ANOVA) of empirical data (i.e., collective recovery time, differences in attack angle of individual predators) were performed in Python 3.8.10 using the 'SciPy' package.

The empirical part of the study is based purely on observational data, with no experimental manipulation, and therefore it is impossible to reproduce exactly. The sample size was maximized from the 19 minutes of video where the positions of predator and prey could be visualised. The attack and evasion patterns of striped marlin and prey are stereotypical, and this suggests that the behaviours recorded in the reported sample represent the behaviours of the striped marlin population being studied. The observed behaviours were superficially confirmed in other videos not used for analysis, however, they were not clear enough to accurately determine predator and prey positions. The same methods can be used each year at the same location where the predator-prey interactions can be observed (provided the predator and prey are present).

The simulation part of the study is reproducible with the model parameters as in Supplementary Table S2 and sample size of $n = 40$ independent simulation realisations. The model sensitivity to the predator-prey speeds, predator attack direction, prey orientational noise and blind angle, as well as the impact of the flee force strength on the collective prey dynamics, were analysed and reported in Supplementary Notes 2–4.

**Reporting summary**

Further information on research design is available in the Nature Portfolio Reporting Summary linked to this article.

**Data availability**

The numerical source data underlying Figs. 1–4 can be found in Supplementary Data 1. The analysed videos[47] from the footage are available on Zenodo with the identifier https://doi.org/10.5281/zenodo.13355844.

The empirical[48] and simulation[49] source data underlying the footage and supporting the findings of this study are available on Zenodo under the identifiers https://zenodo.org/records/13991769 and https://zenodo.org/records/13991999, respectively.

**Code availability**

The code used within this paper is available from the corresponding author upon the request.

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

## Acknowledgements
P.B., J.K., and P.R. acknowledge funding by the Deutsche Forschungsgemeinschaft (DFG, German Research Foundation) under Germany's Excellence Strategy - EXC 2002/1 "Science of Intelligence" - project number 390523135. M.J.H. acknowledges funding by the Deutsche Forschungsgemeinschaft (D.F.G., German Research Foundation) grant - GZ:HA 9403/1-1 AOBJ: 67584. J.E.H-R. was supported by the Whitten Programme in Marine Biology, the Swedish Research Council (2018-04076), and the Office of Naval Research Global (N62909-21-1-2005). Special thanks are due to Max Licht for schematic illustrations of marlin and prey in Fig. 1D and Supplementary Fig. S6; to Korbinian Pacher for assistance in annotating the empirical observations; to Felipe Galván-Magaña at CICI-MAR, La Paz, Mexico, and Magdalena Bay Whale Tours for assistance obtaining videos of striped marlin in the field.

## Author contributions
P.B. and P.R. designed the study; P.B. performed research; M.J.H., F.D., and J.K. acquired the empirical data; P.B. performed the numerical simulations and analysed empirical and simulation results with input from J.E.H-R., P.D., and P.R.; P.B. and J.E.H-R. wrote the paper. All authors commented on the manuscript.

## Funding

## Competing interests
The authors declare no competing interests.
