## [Transparent Peer Review file · Communications Biology]

Collective anti-predator escape manoeuvres through optimal attack and avoidance strategies

Corresponding Author: Dr Palina Bartashevich

This manuscript has been previously reviewed at another journal that is not operating a transparent peer review scheme. The manuscript was considered suitable for publication without further review at Communications Biology.

Version 0:

Reviewer comments:

Reviewer #1

(Remarks to the Author)

In this article, the authors investigate the evasion mechanism known as the "Fountain" effect, which is observed in certain groups of prey when attacked by a predator.

The authors conduct a statistical analysis of experimental trajectories of sardine schools under attack by striped marlins. They examine how the orientation and distance of the fish relative to the predator evolve during the attack and how the school collectively reorganizes to return to its pre-attack state. The authors also propose a modification to an existing agent-based model to include social interactions among the fish and conduct simulations with the aim of understanding how the "Fountain" mechanism is generated.

The authors claim that both the prey's strategies (flee angle) and the predator's strategies (attack angle) are optimized to achieve their respective objectives (escaping/hunting). They show that the optimal values of these parameters strongly depend on the specific metric being optimized (distance to predator/recovery time), and that there is a complex interplay between the dynamics of the predator and the prey. Additionally, they note that the model incorporating social interactions produces the "Fountain" mechanism even in the absence of a blind angle of vision.

I find the data analysis to be technically sound and the results to substantiate the authors' conclusions, making this work of interest to the community.

I have the following points for the authors to consider:

1. In the analysis of the concavity in figure S4, is there any spatial structure of the points corresponding to the "Fountain" effect depending on whether the attack is from the back, front, or side? In other words, can the type of attack be detected based on the concavity of the fish school?
2. The authors define a criterion (LCL) to characterize the recovery time. How robust are the results to variations in this criterion? At what point do the results change qualitatively if the threshold value for defining a return to the initial state is altered?
3. In the experimental results, there are instances where the "Fountain" phenomenon occurs and others where it does not. In the simulations, does this phenomenon always occur? How sensitive is it to the parameter values of the model?
4. In the Main Text, the authors state that the predator moves in a straight line. However, in the Materials and Methods section, they describe the predator as moving linearly toward the updated position of the school centroid. This does not result in a straight-line trajectory, as the predator changes its direction as the fish school moves (which occurs due to its high polarization). A brief clarification on this point would help avoid misunderstandings.
5. In figure S7, the results indicate that including a blind angle does not have uniform consequences across all scenarios, contrary to the authors' claims. In the fourth row, which corresponds to a blind angle of $2(3\pi/12)$, side attacks behave very differently from frontal attacks in the absence of noise ($D=0$). Conversely, in the case without a blind angle, the behavior is nearly identical between both scenarios for $D=0$. Do the authors have any hypotheses for this discrepancy? Could it be related to the interplay between the initial orientation of the prey and the predator (front, back, side) and the location of the blind angle?

6. Studies in human groups, such as Physical Review Letters 113.23 (2014): 238701, suggest that the critical factor is not the distance to the predator (or collision in that case), but the time until the predator reaches the prey (or collision occurs). This could play a significant role in the differences between front, back, and side attacks and the respective strategies. I suggest that the authors explore this idea, even briefly, and discuss how it relates to their findings.

Reviewer #2

(Remarks to the Author)

In this manuscript the authors studied the details of a fountain escape response in sardines being attacked by striped marlins. In their study the authors used drone footage, and manual annotation of the marlins and sardines, to characterize location and orientation of the predator during the attack and of a randomly selected number of prey for some of the frames of the footage. The authors managed to get a large amount of predation attempts ($n > 100$), of which a selected few were used in the specific fountain statistics ($n = 16$), impressive numbers considering they were acquired in the field. The authors then compare their experimental results with an adaptation of a previously developed model to better characterize and understand why side attacks were the most common, and if prey reaction to them was being optimized.

I truly enjoyed this study, but some small corrections are needed in order to extract the best value of it. Here are some of my recommendations:

When analyzing the change of polarization of the school under attack, the authors synchronized the polarization time series of each attack by their lowest polarization value, an appropriate choice. Nevertheless, some information should be given on what was the range of shifts they had to introduce to present the figures as seen in Fig. 1D. I imagine a boxplot similar to Fig. 1D, but placed in the SI material should suffice. Another thing that I would enjoy seeing is for the authors to present error bars of the shaded areas in Fig. 1D. Figures 1B and 1C should also contain the total number of events for each case in its title.

Figure 2B labels are very hard to see, and in the PDF we received the figure isn't in a vectorial format, therefore it's quite hard to zoom in and read it appropriately.

Figure 3 can be completely removed in my opinion, or at least moved to SI. It is very hard to understand it, and the author's only mention of it is in its very brief subsection "impact of the flee angle on self-organised dynamics", and its equally brief SI description. Fig 3A and 3C have no units for the radial axis, which I understand is a normalized metric, but I would say trying to understand these two figures took me most of my time during this review. I understand that probably the main point of the figure is to show the overall good agreement of simulations with experimental data, but then the authors failed to actually use it and describe it accordingly later on the text, so it just feels like a placeholder. Also when searching for mentions of Fig. 3, I noticed that the authors mixed definitions of referencing figures, e.g Fig. 3 vs Fig 3. Consistency is appreciated for these cases when a reader wants to quickly find all references to a specific figure.

I overall enjoyed their modeling effort and the author's overall exploration of it. Nevertheless, one of the biggest claims of the authors is that this model differs from previous works, by having the social forces (alignment and attraction), still being active when the fish are performing their flee event. While looking at the SI material I was quite surprised to see that the authors did a parameter scan on how noise and blind angle would affect the model results, but then failed to explore how different ratios of social forces vs flee intensity would affect their results. I understand that in general the flee intensity should be higher than the social forces, but the paper used a single set of values for this (flee intensity of 50, while both social forces being equal to 3). Considering the author's claims, I would expect a few different ratios of these forces being used to justify social forces are indeed meaningful and their results are: robust in different ranges; and/or finding that only very high ratios still produce this behavior, therefore previous works were indeed correct into excluding social forces; showing how no social forces would change (or not) their results. Only a brief mention in the text and SI figures should suffice, unless their findings drastically change the understanding of the work, although I doubt this would be the case.

I found figure 4 to be quite well done and explanatory of the point they were making, but I would like to see the same x and y axis in for 4A-C and perhaps then remove 4D?

The authors mention that they performed manual annotation, but also that they used a custom matlab code to extract escape angles. I'm not sure of this specific journal, but a comment on availability or "code available upon request" is often standard.

Some of the SI material I found a bit lacking, in specific the examples of attack events and their tracking Fig S2 and S3. First, given that the authors only analyzed 16 attack events with fountain escapes, a video of all these events should be added here. Additionally, for the figure itself, a grid of rows representing each attack, and columns representing the "start time, end time, lowest polarization and recover time, time should be added to each example shown. Arrow heads representing fish should have the same size, since they don't represent speed, and only their stem varies its length (given by the manual annotation of head vs tail).

Reviewer #3

(Remarks to the Author)

The paper is well-written and only requires minor revisions. It represents a significant study on predator-prey interactions in the wild, which is remarkable given the challenges of such a study system and method. The authors analyze the predator

attack strategy and prey escape behavior of striped marlin and sardine schools using drone footage as well as an agent-based model. The results are clear, well-analyzed, and well-described.

Here are my comments:

- Figure 1: For figure 1B, consider adding the total number of attacks in the legend ($N=104$) to avoid confusion with subsequent analyses. Similarly, add the number of attacks in Figure 1C, as the total number of videos included in following analysis is reduced. I noticed the information is included in the caption but I would still recommend adding it to the figures.

- Figure 1D: The figure is informative, but I suggest adding information about the shaded bar. To illustrate the variation in start/end times of different trials, consider creating a separate boxplot with all the data points, or adding an error bar around the shaded bar (the extra figure could be in the supplementary material). It would be helpful to know the number of frames each line was shifted after aligning to the minimum polarization value. Also, always for figure 1D, I would recommend making the lines of each encounter (grey lines) more visible.

- Time Interval Between Attacks: Are there instances where the same predator attacks more than once within a short time interval? If so, what is the average time interval between attacks by the same individual? The supplementary material mentions one case where the fish school did not recover before the next attack. Does this occur in other attacks, even if they are not included in the analysis? Can any information be extracted about the attack strategy, whether it is the first attack or a subsequent one by the same predator (e.g. side/front/back)? Although it may not always be possible due to image quality, is it sometimes possible to determine if a hunt attempt was successful?

- Figure 2: Considering that Figure 1 reports the recovery time and Figure 2C reports the distance to a predator, could you add to Figure 2 the equivalent of Figure 4D but with empirical data instead of simulation data? I find Figure 4D very interesting for understanding the optimal strategies of both predator and preys and I would expect a similar pattern for empirical data.

- Figure 3: I would suggest removing this figure and moving it to the supplementary material. I find it hard to understand due to lacking figure labels (e.g., y-label of plots A and C), making it difficult to interpret. Moreover, the figure is mentioned only once in the main text, so it might be better suited for the supplementary material. Alternatively, you could improve it by fixing the figure axes labels and descriptions in the main text. I am unsure if I understand what plots A and C are illustrating and which data is selected for the plot.

- Figure 4: I would adjust both x and y axes to use the same range for all figures A-D. This way, I find that it could possibly be easier to compare the recovery times associated with each attack type front, back, side.

- Figure S1: 8 predators were subsampled with 4 attacks each. I suggest highlighting in this figure the 16 attacks (if present) also analyzed in the main text. Are these 8 predators also part of the 11 predators of Figure 1D? (I am referring to information written in the second paragraph of the supplementary material). Why were some predators excluded from the ANOVA test?

- Figure S2: Given that Figure 1D includes 16 attacks, would it be possible to include one screenshot per attack as supplementary material? Alternatively, may be better to explain how the 6 instances were selected. What is the information contained in the size of the yellow arrow? If it is not relevant, consider keeping the size equal for each individual. Also, how was the frame of the instance decided? Is it an analogous moment across attacks? For instance, considering all trials were shifted based on the moment of lowest polarization of the group, could the instances in Figure S2 all be from a similar time in the attack?

- Video material: Could you share the videos of the selected attacks along with the publication? This is not necessary but could be an interesting addition.

Overall, the paper is a significant contribution to the field and these minor revisions may improve its clarity.

Reviewer #4

(Remarks to the Author)

In this article, the authors investigate the evasion mechanism known as the "Fountain" effect, which is observed in certain groups of prey when attacked by a predator.

The authors conduct a statistical analysis of experimental trajectories of sardine schools under attack by striped marlins. They examine how the orientation and distance of the fish relative to the predator evolve during the attack and how the school collectively reorganizes to return to its pre-attack state. The authors also propose a modification to an existing agent-based model to include social interactions among the fish and conduct simulations with the aim of understanding how the "Fountain" mechanism is generated.

The authors claim that both the prey's strategies (flee angle) and the predator's strategies (attack angle) are optimized to achieve their respective objectives (escaping/hunting). They show that the optimal values of these parameters strongly depend on the specific metric being optimized (distance to predator/recovery time), and that there is a complex interplay between the dynamics of the predator and the prey. Additionally, they note that the model incorporating social interactions produces the "Fountain" mechanism even in the absence of a blind angle of vision.

I find the data analysis to be technically sound and the results to substantiate the authors' conclusions, making this work of

interest to the community.

I have the following points for the authors to consider:

1. In the analysis of the concavity in figure S4, is there any spatial structure of the points corresponding to the "Fountain" effect depending on whether the attack is from the back, front, or side? In other words, can the type of attack be detected based on the concavity of the fish school?
2. The authors define a criterion (LCL) to characterize the recovery time. How robust are the results to variations in this criterion? At what point do the results change qualitatively if the threshold value for defining a return to the initial state is altered?
3. In the experimental results, there are instances where the "Fountain" phenomenon occurs and others where it does not. In the simulations, does this phenomenon always occur? How sensitive is it to the parameter values of the model?
4. In the Main Text, the authors state that the predator moves in a straight line. However, in the Materials and Methods section, they describe the predator as moving linearly toward the updated position of the school centroid. This does not result in a straight-line trajectory, as the predator changes its direction as the fish school moves (which occurs due to its high polarization). A brief clarification on this point would help avoid misunderstandings.
5. In figure S7, the results indicate that including a blind angle does not have uniform consequences across all scenarios, contrary to the authors' claims. In the fourth row, which corresponds to a blind angle of $2(3\pi/12)$, side attacks behave very differently from frontal attacks in the absence of noise ($D=0$). Conversely, in the case without a blind angle, the behavior is nearly identical between both scenarios for $D=0$. Do the authors have any hypotheses for this discrepancy? Could it be related to the interplay between the initial orientation of the prey and the predator (front, back, side) and the location of the blind angle?
6. Studies in human groups, such as Physical Review Letters 113.23 (2014): 238701, suggest that the critical factor is not the distance to the predator (or collision in that case), but the time until the predator reaches the prey (or collision occurs). This could play a significant role in the differences between front, back, and side attacks and the respective strategies. I suggest that the authors explore this idea, even briefly, and discuss how it relates to their findings.

Version 1:

Reviewer comments:

Reviewer #2

(Remarks to the Author)

I am happy to see that the authors have addressed all reviewer comments and requests. I was especially happy to see that the authors have accepted my suggestions and added additional simulations, which have now cemented the conditions that cause the fountain effect in self propelled particle models. I was also glad to see that Figure 3 is now properly referenced and justified in much more detail than before.

Despite being unusual in my experience, after the first round of reviews the authors have made several figures for the reviewers only, and added their reasoning on why these figures didn't have to be included in the text. I do overall agree (at least for the ones suggested by me), that these "reviewers only" figures don't bring sufficient improvement to the manuscript to justify adding them.

I am quite satisfied with all the changes, and I think that the article is even better with them.

Reviewer #3

(Remarks to the Author)

I have reviewed the revised manuscript and appreciate the efforts made to address the feedback provided in the previous round of revisions. All of the points I raised have been satisfactorily addressed, and I have no further requests or concerns.

Reviewer #4

(Remarks to the Author)

In this resubmission of the manuscript titled "Collective Anti-Predator Escape Manoeuvres Through Optimal Attack and Avoidance Strategies," the authors have thoroughly and positively addressed all of the concerns I previously raised.

The revisions, both in response to my feedback and that of the other reviewers, have greatly improved the overall quality of the manuscript. Consequently, I am pleased to recommend it for publication in Communications Biology.

Response to Reviewing on Manuscript COMMSBIO-24-2603-T

Comments from Reviewer #1/#4

Comment 1. In the analysis of the concavity in figure S4, is there any spatial structure of the points corresponding to the "Fountain" effect depending on whether the attack is from the back, front, or side? In other words, can the type of attack be detected based on the concavity of the fish school?

Response: We thank the reviewer for raising this interesting question. We want to emphasize that the primary purpose of Fig. S4 is to demonstrate that the concavity of the fish school allows us to quantitatively distinguish between the "fountain" and "non-fountain" responses. This way, Fig. S4 serves as illustrative evidence for objective, quantified classification confirming our visual pattern classification, based on the descriptive definition of the "fountain effect". Regarding classifying the "fountain" effect further among the attack types, we can expect that different angles of attack will result in slightly different collective escape patterns. We have updated Fig. S4 (corresponding to Fig. S6 in the revision) by coloring the "fountain" cases (denoted by dots) for the attack direction from the back (b, in blue), front (f, in red), and side (d, in green). While "fountain" responses triggered by the back attacks are clustered (except b_c), the ones by side and front attacks are more scattered, making it hard to assign them to a particular class (see Fig. S4). Therefore, while the concavity metric allowed us to distinguish and select between the "fountain" and "non-fountain" responses, the concavity metric alone may be insufficient for making a proper distinction between attack types or more data will be needed to validate such a classification. However, it remains unclear how general or robust the distinguishing features will be with respect to other core parameters like fish school size, relative speeds, etc. Finally, while such type of classification can be potentially very interesting in different contexts, it does not add to addressing the research questions of our manuscript. Nevertheless, we added the updated Fig. S4 (corresponds to Fig. S6 in the revision), with the coloration of the data points depending on the attack direction, into Supplementary Materials of the revised version of the manuscript.

Figure S6. Each dot shows the convexity of the polygon defining the borders of the prey school geometry in the middle C_m and in the end C_e of an attack over annotated $n=37$ non-fountain (crosses) and $n=14$ fountain (dots) evasions. The middle of an attack is defined as an approximate point in time between the start and the end of an attack. The start of the attack is defined as 3 frames before the predator's bill is on one level with the prey fish, such that a drawn perpendicular line to the bill's tip intersects at least with one sardine. The end is defined as when there is no more any fish in front of the mouth and the bill of the predator. The capital letters correspond to the non-fountain evasions in Fig. S5 and the lower case letters indicate the attack directions of the respective fountain evasions (denoted by the subscript letters) as in Figs. S2-S4. The attack directions are denoted by colors and lowercase letters, such as attack from the back (b in blue), side (s in green), and front (f in red).

Comment 2. The authors define a criterion (LCL) to characterize the recovery time. How robust are the results to variations in this criterion? At what point do the results change qualitatively if the threshold value for defining a return to the initial state is altered?

Response: We believe the results are robust to the introduced criterion (LCL) as the estimation of collective recovery time is based (1) on the baseline order state of the prey school Φ_0 without a predator estimated over 40 simulations (see Fig. R1-a), thereby allowing us to account for the variance due to the internal noise; and (2) we applied Savitzky-Golay filter on top of the obtained time-series to ensure the robustness. Below, we provide examples illustrating the computation of the defined LCL criterion, depending on different threshold values. The threshold value for a return to the initial state is defined as follows $LCL := \bar{\Phi} - 3\bar{\sigma}$. To vary the threshold one can use different levels of standard deviation (std) of the baseline signal $\bar{\sigma}$. Fig. R1 (a) shows the time-averaged mean polarisation $\bar{\Phi}$ of the prey group denoted by dashed red line and estimated over 40 simulations (grey lines) along with respective levels of std $\bar{\sigma}$. As mentioned in the SI of the current manuscript, the estimated recovery time is defined by the timestep t when $\hat{y}(t) = 0$, where $\hat{y}(t)$ is a smoothed by Savitzky-Golay filter time series difference between polarisation timeseries of the perturbed prey and LCL. Figs. R1 (b,d,f) show how estimated recovery times (i.e., $\hat{y}(t) = 0$) vary depending on different levels of std selected in LCL (i.e., $\bar{\Phi} - 3\bar{\sigma}$, $\bar{\Phi} - \bar{\sigma}$, $\bar{\Phi} - 2\bar{\sigma}$, $\bar{\Phi} - 5\bar{\sigma}$). Figs. R1 (c,e,g) show how the estimated recovery times, depending on different threshold values, relate to the value of polarisation of the prey. As we can see from Figs. R1 (c,e,g), there will be no qualitative difference depending on the std threshold, particularly between the ones below $3\bar{\sigma}$, as they all quantitative correspond to high levels of polarisations $\Phi \geq 0.9$, which is indicative of an ordered group state. In other words, the results change qualitatively at the point below the defined threshold by LCL with $3\bar{\sigma}$. Also, Figs. R1 (c,e,g) illustrates that choice of $3\bar{\sigma}$ is more robust to small perturbations in the signal compared to $5\bar{\sigma}$.

We have added Fig. R1 as Fig. S13 and the description as above in lines 857-863 of the revised version of the Supplementary Material.

Figure R1. (a) Time series of baseline polarization values of the non-perturbed prey group over 40 simulations (in grey) with a respective time-averaged mean value (dashed line in red) along with different standard

deviation levels. (b,d,f) Instances of the smoothed time-series difference of the polarization values between the perturbed prey group and the mean value of the non-perturbed one as in (a). The dots correspond to the estimated collective recovery time depending on the selected in (a) level of standard deviation ($3\bar{\sigma}$ in black, $\bar{\sigma}$ in green, $2\bar{\sigma}$ in orange, $5\bar{\sigma}$ in blue). (c,e,f) Instances of the time series of polarization values for the perturbed prey group with marked recovery times (by dots) depending on the level of selected standard deviation in (b,d,f) respectively.

Comment 3. In the experimental results, there are instances where the "Fountain" phenomenon occurs and others where it does not. In the simulations, does this phenomenon always occur? How sensitive is it to the parameter values of the model?

Response: Our model is based on the previous work (Klamser and Romanczuk 2021) and is specifically adjusted to simulate the "fountain" phenomenon. The occurrence of the "fountain effect" primarily depends on the model parameters such as the individual prey flee angle, distance-regulating force, and the predator attack strategy. In response to Reviewer 2's suggestion, we added an analysis of the effect of the flee force ratio to social forces in the revised Supplementary Materials, lines 955-961, and Fig. S10. While keeping the model parameters that lead to the "fountain effect" unchanged, non-fountain responses can also be generated by altering the predator's attack strategy. For example, in the simulation, introducing an angular shift of 0.15π on the detected centre of mass (COM) of the prey group from the predator's perspective leads to the escape of the prey group on one side without splitting, avoiding the predator altogether (see Video 1, direct download link for reviewers: <https://box.hu-berlin.de/f/9ce51ed5812743afa2f4/?dl=1>). However, our primary focus was on instances where the predator attacked by aiming at the COM of the prey group. This approach always triggers a "fountain" response in the simulation, as also seen in empirical observations, making it possible for a direct comparison with empirical data.

Comment 4. In the Main Text, the authors state that the predator moves in a straight line. However, in the Materials and Methods section, they describe the predator as moving linearly toward the updated position of the school centroid. This does not result in a straight-line trajectory, as the predator changes its direction as the fish school moves (which occurs due to its high polarization). A brief clarification on this point would help avoid misunderstandings.

Response: Thank you very much for pointing out at these two contradicting statements. The confusion arises because the predator initially aims at the centre of mass (COM) of the moving school, but as it reaches the COM, it resumes straight motion at its current velocity. Interestingly, this results into almost perfectly straight-line trajectory over the duration of the "fountain effect" for front and back attacks, while for side attacks it is not always the case (see Fig. R2).

We have introduced the following clarifications in the revised version of the manuscript:

In the main text (lines 297-300): "Independent of the attack angle, we consider that the predator is moving linearly towards the updated position of the prey's school centroid and switches to moving straight with its current velocity \vec{v}_p when it enters the ϵ - neighbourhood of \vec{r}_{com} ($\epsilon := |\vec{r}_{com} - \vec{r}_p| \leq 1$)."

In the Supplementary Materials (lines 835-837): "The predator moves linearly to the updated position of the prey's group centroid \vec{r}_{com} at each simulation step without noise in its velocity orientation \vec{v}_p . Once predator gets into the vicinity of the preys' centroid, i.e., $\epsilon := |\vec{r}_{com} - \vec{r}_p| \leq 1$, it continues moving straight at its current speed resulting in a straight-line trajectory after passing \vec{r}_{com} ."

Figure R2. (a)-(c) Instances of the predator's (shades of red) and the COM (shades of blue) trajectories over the duration of the "fountain effect" for back, front, and side attacks. Green and pink dots (stars) indicate the start and end of the predator's (COM) trajectory, respectively. The red cross indicates the moment on the predator's trajectory when it gets into the vicinity of COM (denoted by the red star for the COM's trajectory). (d) The straightness index D/L of the predator's trajectories in the simulation based on the net displacement D of the predator's position at the start of the "fountain effect" and at its end, i.e., $D := |\vec{r}_p(t_{end}^{fnt}) - \vec{r}_p(t_{start}^{fnt})|$, and the actual path length travelled L .

Comment 5. In figure S7, the results indicate that including a blind angle does not have uniform consequences across all scenarios, contrary to the authors' claims. In the fourth row, which corresponds to a blind angle of $2(3\pi/12)$, side attacks behave very differently from frontal attacks in the absence of noise ($D=0$). Conversely, in the case without a blind angle, the behavior is nearly identical between both scenarios for $D=0$. Do the authors have any hypotheses for this discrepancy? Could it be related to the interplay between the initial orientation of the prey and the predator (front, back, side) and the location of the blind angle?

Response: Yes, the blind angles does not have uniform consequences across all mentioned scenarios, as can be seen from Fig. S7 (Fig. S11 in the revised version). We have specified in the revised manuscript the following claims for better clarity:

In the main text (lines 363-368): "With an additional blind angle on top of the Voronoi neighbourhood, the optimal flee angle remains robust relative to the shift imposed by the blind angle in the range of $[15^\circ, 30^\circ]$, i.e., the resulting flee angle is the sum of the blind angle and the $\Delta\alpha_{flee}^*$ (see SI Appendix, Fig. S11)."

In the Supplementary Materials (lines 933-935): "The performance for back attack relative to front and side attacks is robust at various noise levels D_ϕ in the absence of an additional prey blind angle (the first row of Fig. S11)."

In the Supplementary Materials (lines 937-940): "To note, $D_\phi = 0.2$ remains optimal in terms of $\langle r_{ip} \rangle$ regardless of the prey's blind angle within its feasible range (i.e., $\Delta\alpha_{blind} \in [15^\circ, 30^\circ]$). For $D_\phi = 0.2$, with an additional prey blind angle $\Delta\alpha_{blind}$ on top of the Voronoi tessellation, the optimal flee angle $\Delta\alpha_{flee}^* = 30^\circ$ remains robust in case of the front attack and shifts to $\Delta\alpha_{flee}^* + \Delta\alpha_{blind}$ in case of back and side attacks for $\Delta\alpha_{blind} \in [15^\circ, 30^\circ]$ (see Fig. S11)."

To address the noted by the reviewer difference between front and side attacks in the case of $D_\phi = 0$ for $\Delta\alpha_{blind} = 45^\circ$, we have also added the following paragraph in the Supplementary Materials

(lines 941-954): "The interplay between the initial prey orientation, the size of its blind zone, and the predator's relative position to the prey defines whether the prey will actively flee from the predator (i.e., be a direct responder) or not. This, in turn, impacts the further social propagation of the response to the neighbours, affecting the collective response overall. In this regard, the front attack is affected the least by the presence of the blind angle, as the blind zone behind the prey does not alter the perception of the predator in front of the prey. Fig. S11 ($D_\phi = 0$) supports this claim, as the performance for the front attack does not change regardless of the blind angle. The cases of side and back attack differ from the front one, since there the size of the blind zone affects the perception of the predator and, hence, whether the fleeing force is activated or not. In particular, the side attack is characterised by a greater variability in detecting individuals at the edge of the school depending on their orientation, compared to an attack from behind. Moreover, the spread of the response is further impaired by the inability to detect responding prey individuals, since they may be in the blind spot of their neighbours. This results in a non-linear dependency for the distance away from the predator which the prey can build, depending on the flee angle for $\Delta\alpha_{blind} \in [45^\circ, 60^\circ]$, particularly in the case of $D_\phi = 0$. The introduction of the noise on the prey orientation allows for oscillations in the position of the blind zone, creating a chance for better social propagation of the response, thereby smoothing the dependence (see $D_\phi > 0$, $\Delta\alpha_{blind} \in [45^\circ, 60^\circ]$, side attack)."

Comment 6. Studies in human groups, such as Physical Review Letters 113.23 (2014): 238701, suggest that the critical factor is not the distance to the predator (or collision in that case), but the time until the predator reaches the prey (or collision occurs). This could play a significant role in the differences between front, back, and side attacks and the respective strategies. I suggest that the authors explore this idea, even briefly, and discuss how it relates to their findings.

Response: While this is an interesting avenue of thinking, we do not have the ability to test it at the moment due to deficiencies in the quality of spatial-temporal data. Calculating the time-to-collision knowing the distance between each individual prey and the predator, as well as the prey's velocity relative to the predator. Unfortunately, we lack the detailed trajectory data necessary to accurately estimate the relative velocities of individual prey for different attack directions in our experimental data. However, using the available data (see Fig. 3A in the manuscript)—which includes the distance from the predator to a randomly selected individual prey every third frame, along with their velocity orientation (but not the magnitude)—and a general estimate of the predator-to-prey speed ratio (i.e., 2:1), we computed the following probability density plots (see Fig. R3):

Figure R3. Pair probability density plot of the occurrences of predator-prey pairs being at a certain distance versus them being at a certain time to collision (for predator and randomly selected prey individuals during the attack).

While the probability densities of time-to-collision for side and front attacks are relatively similar, the back attack shows a distinct difference. However, due to the limited data and the rough estimation of speeds, we are cautious about drawing strong conclusions from these plots and will not include them in the manuscript. But we have added the following paragraph in the discussion section of the revised manuscript (lines 575-585): "Prey may also prioritise distance by maximising the time to collision between themselves and predators. Some studies in human crowds (Karamouzas et al, 2014) suggest that time-to collision may play an important role in the social interaction, which is equivalent to relative speed dependent repulsion (Romanczuk et al, 2009), and that these type of interactions may play a role in predator evasion. However, these interactions have so far not been considered with a fleeing angle, and it is unclear under what conditions, or whether at all, they can produce a robust fountain effect. This should be addressed by future research."

Comments from Reviewer #2

Comment 1. When analyzing the change of polarization of the school under attack, the authors synchronized the polarization time series of each attack by their lowest polarization value, an appropriate choice. Nevertheless, some information should be given on what was the range of shifts they had to introduce to present the figures as seen in Fig. 1D. I imagine a boxplot similar to Fig. 1D, but placed in the SI material should suffice.

Response: Thank you for acknowledging our approach. The time shifts were based on random video cuts of the attack events from longer footage and therefore do not convey any scientific information. These time shifts were applied primarily for 'cosmetic' purposes, to improve the visibility of overlapping plot lines and to highlight the similarity in polarization patterns across instances of the same attack type. As suggested by the reviewer, we created a figure (see Fig. R4 below) with box plots showing the range of shifts introduced in Fig. 1D. However, we believe it is unnecessary to include this in the Supplementary Materials for the reasons mentioned above. Additionally, the amount of shift is already visible in Fig. 1D, as not all plot lines start from time zero. Nevertheless, we have mentioned the amount of time shift in the newly added Table S3 in the revised version of the SI.

Figure R4. Time-shifts on the original data to present the polarization over time as seen in Fig. 1D (0.01s = 3 frames).

Comment 2. Another thing that I would enjoy seeing is for the authors to present error bars of the shaded areas in Fig. 1D.

Response: Done as suggested by the reviewer. We added the error bars for the "fountain" start and end times in Fig. 1D (see below) defining the shaded areas. Additionally, we add the boxplots of the duration of the "fountain" evasion in the Supplementary Materials as Fig. S16 for better tractability of the introduced error bars in Fig. 1D.

Figure 1D. The error bars at the borders of the windows indicate the standard deviation in the "start" and "end" times of the fountains (for more details see Table S3, Fig. S15)

Figure S16. Duration of the "fountain effect" depending on the attack direction (front, back, and side). According to the Kruskal-Wallis test followed by the Tukey test for multiple pairwise comparison, the results are non-significantly different (ns).

Comment 3. Figures 1B and 1C should also contain the total number of events for each case in its title.

Response: Done as suggested by the reviewer. We have added the number of events for each case ($n=104$) and ($n=67$) in the titles of Fig. 1B and Fig. 1C, respectively.

Comment 4. Figure 2B labels are very hard to see, and in the PDF we received the figure isn't in a vectorial format, therefore it's quite hard to zoom in and read it appropriately.

Response: Done as suggested by the reviewer. Fig. 2 inside the revised version of the manuscript is in a vectorial format, and we increased the size of the labels in that figure.

Comment 5. Figure 3 can be completely removed in my opinion, or at least moved to SI. It is very hard to understand it, and the author's only mention of it is in its very brief subsection "impact of the flee angle on self-organised dynamics", and its equally brief SI description. Fig 3A and 3C have no units for the radial axis, which I understand is a normalized metric, but I would say trying to understand these two figures took me most of my time during this review. I understand that probably the main point of the figure is to show the overall good agreement of simulations with experimental data, but then the authors failed to actually use it and describe it accordingly later on the text, so it just feels like a placeholder.

Response: Yes, one of the main points of Fig. 3 is to illustrate the available empirical data as well as its good agreement with simulations. Fig. 3 is inspired by the previous work of Hall et al. (1986), where similar plots were created in Figures 5 and 7 using empirical data obtained in a laboratory conditions, in a tank. In their study, a ball was used to imitate a predator attacking from the back of the fish school, triggering the "fountain" response, while other attack directions remained unstudied. This way, another point of our Figs. 3A,B was also to show the good agreement of our empirical data from the field with the one obtained in the lab by Hall et. (Fig. 5), for the case of the back attack. Moreover, Figs. 3A,B illustrate the observed escape angles of the prey in the wild, which is a primary focus of the study in the simulations. For these reasons, we believe that Fig. 3 should remain in the main text. We have added a label "dist. to a pred., d_{ip} " with respective units for the radial axes of Figs.3A,C, where the simulation data represent dimensionless (random) units (denoted as r.u.). We have also added a schematic illustration of the marlin at the origin of the polar plots to enhance interpretability. Additionally, we incorporated the description of Fig. 3, which was previously in the SI section "Polar plot illustration of fountain escape dynamics", into the main text (lines 406-425) of the revised manuscript and provided a more detailed explanation of the plot results in lines 426-449.

Lines 406-427: "Fig. 3A shows the observed evasion reactions over the duration of the "fountain effect" plotted relative to the predator (the origin of the coordinate system) for all analysed encounters and each attack direction. For each attack encounter, 20 random fish were annotated each third frame up until the predator left the school. Each arrow in Fig. 3A represents the average position and orientation of the fish in each encounter at 10° increments of θ_1 relative to the

predator. The amount of arrows in each 10° angular sector is defined by the amount of respective attacks. This way, Fig. 3A illustrates the aggregation of all attacks of each type, highlighting the similarity of the encounters in their response. To ensure a fair visual illustration between simulation and empirical data (Fig. 3C vs. Fig. 3A), we subsampled the simulation data to match the amount of empirical data. That is, in Fig. 3C 20 prey-agents were uniformly selected at random from the prey group at each iteration over "fountain" duration and the same number of attacks for each type as in the empirical data was used. Overall, Figs. 3A,C describe the average escape angles over $t_w := [t_{start}^{fnt}, t_{end}^{fnt}]$ as the predator passes through and exits the school."

Lines 428-451: "Fig. 3B shows that, in case of the back attack, for $\theta_1 \in [0^\circ, 100^\circ]$ there is a consistency in the prey escape angle around $\theta_2 \approx 150^\circ$. As the prey get into the zone behind the predator $\theta_1 > 100^\circ$, there is a sharp decline in θ_2 indicating that the fish are turning to reunite with the other half of the school formed during the split. These results are consistent with previous work by Hall et al. (1986) performed in the laboratory conditions. Since the fish orientation and positions are computed relative to the predator, the results vary depending on the attack direction due to the predator's movement and its initial relative position at the start of the attack. Front attacks are characterised by the sharp increase in θ_2 from approx. 25° up to 150° for the individuals in front of the predator $\theta_1 \in [0^\circ, 75^\circ]$. The escape angle levels up at around 150° for $\theta_1 \in [75^\circ, 125^\circ]$ as the predator gets inside the school and declines for $\theta_1 > 125^\circ$ as the prey reunite behind the predator's tail. Side attacks are described by greater variability across the encounters, compared to back and front attacks. Similarly to front attacks, for $\theta_1 \in [75^\circ, 125^\circ]$ the average observed prey escape angles in the case of side attacks vary around $\theta_2 \approx 150^\circ$. Figs. 3C,D demonstrate that incorporation of $\Delta\alpha_{flee} = 30^\circ$ into the model qualitatively reproduced collective patterns of escape in the simulation as observed in the empirical data (Figs. 3A,B)."

Comment 6. Also when searching for mentions of Fig. 3, I noticed that the authors mixed definitions of referencing figures, e.g. Fig. 3 vs Fig 3. Consistency is appreciated for these cases when a reader wants to quickly find all references to a specific figure.

Response: Thank you for noticing this. The following references were fixed: line 261 (Fig. 2A), line 268 (Fig. 2A), line 281 (Fig. 2B), line 311 (Fig. 2A), line 318 (Fig. 2B); in SI: line 778 (Fig. 1A).

Comment 7. I overall enjoyed their modeling effort and the author's overall exploration of it. Nevertheless, one of the biggest claims of the authors is that this model differs from previous works, by having the social forces (alignment and attraction), still being active when the fish are performing their flee event. While looking at the SI material I was quite surprised to see that the authors did a parameter scan on how noise and blind angle would affect the model results, but then failed to explore how different ratios of social forces vs flee intensity would affect their results. I understand that in general the flee intensity should be higher than the social forces, but the paper used a single set of values for this (flee intensity of 50, while both social forces being equal to 3). Considering the author's claims, I would expect a few different ratios of these forces being used to justify social forces are indeed meaningful and their results are: robust in different ranges; and/or finding that only very high ratios still produce this behavior, therefore previous works were indeed correct into excluding social forces; showing how no social forces would change (or not) their results. Only a brief mention in the text and SI figures should suffice, unless their findings drastically change the understanding of the work, although I doubt this would be the case.

Response: Thank you for your positive feedback and for recognising our modelling effort. We are glad that you enjoyed our exploration and analysis, and we appreciate your suggestion. The force ratio was chosen to qualitatively reproduce an observed "fountain"-like pattern and primarily to distinguish it from a "vacuole"-type response. In a "vacuole" response, the predator is surrounded by prey from all sides as they create a gap in the centre of the school. For the revised manuscript, we conducted additional experiments by varying the flee strength intensity $\mu_{flee} \in \{6, 10, 20, 30, 40\}$ while keeping the social force intensity fixed at $k = \mu_{flee} = 3$. The results, shown in Fig. S10 (similar to Fig. 3D in the main text), indicate that high flee intensities ($\mu_{flee} > 20$) produce outcomes closest to the empirical data (see Fig. 3B in the main text). We have also added snapshots from the simulation in Fig. S9, showing that lower flee strength ($\mu_{flee} < 20$) leads to a "vacuole" response, where prey behind the predator reunite while the predator is still within the school. This suggests that a "fountain effect" only occurs with sufficiently strong flee strength, where social forces become negligible for directly responding individuals.

We have revised certain claims in the main text, lines 304-305, 314-318, which now read as follows: "We consider social interactions between prey which are maintained during predator avoidance. However, the "fountain effect" is primarily observed when the flee strength μ_{flee} is sufficiently strong (see SI Appendix, Figs. S9, S10), such that social forces become negligible for directly responding individuals, similar to previous models, e.g., (11, 27)."

We have also added a brief section "Effect of varying flee force strength on collective response" in the Supplementary Materials in lines 955-961, featuring the newly added Fig. S9 and Fig. S10: "We explored the impact of the flee force to social force intensity ratio on prey self-organised dynamics by conducting simulation experiments with varying flee strength intensity $\mu_{flee} \in \{6, 10, 20, 30, 40\}$, while keeping the social force intensity fixed at $k = \mu_{flee} = 3$ with $\Delta\alpha_{flee}^* = 30^\circ$ and $D_\varphi = 0.2$. The results, shown in Fig. S10 (similar to Fig. 3D in the main text), indicate that high flee intensities $\mu_{flee} > 20$ produce patterns closest to the empirical data (see Fig. 3B in the main text). This suggests that a "fountain effect" occurs primarily with sufficiently strong flee strength, while lower flee intensity results in a more "vacuole" type of collective response, where the predator is surrounded by the prey from all sides while in the middle of the prey school (Fig. S9)."

Figure S10. Effect of varying flee force strength $\mu_{flee} \in \{6, 10, 20, 30, 40\}$ on the self-organised dynamics of the prey response in the simulation for $k = \mu_{flee} = 3$ and $D_\varphi = 0.2$. Relationship between prey's position angle relative to the predator (θ_1) and prey's orientation (swimming) angle (θ_2), given $\Delta\alpha_{flee}^* = 30^\circ$. Each grey line represents a single simulation run, while the thick line in colour shows the average over the instances. The predator:prey speed ratio is set to 2:1.

Figure S9. Snapshots of the simulation in the middle of the attack, when the predator (red circle) attacked from the back of the prey school (blue circles), depending on the flee force intensity $\mu_{flee} \in \{6, 10, 20, 30, 40\}$, given $k = \mu_{flee} = 3$, $D_\varphi = 0.2$ and $\Delta\alpha_{flee}^* = 30^\circ$. Lower flee strength ($\mu_{flee} < 20$) results in a “vacuole” response, where prey behind the predator reunite while the predator is still within the school. In contrast, stronger flee force ($\mu_{flee} \geq 20$) leads to “fountain”-like patterns. The predator:prey speed ratio is set to 2:1.

Comment 8. I found figure 4 to be quite well done and explanatory of the point they were making, but I would like to see the same x and y axis in for 4A-C and perhaps then remove 4D?

Response: We are glad to hear that you found Fig. 4 well-explanatory, as it is the central figure of our manuscript. While preparing our first version of the manuscript, we considered having the same x and y axes for Fig. 4A-C, however, we did not find them appealing as they made it harder to focus on the outcomes. We attach Fig. R6 below for your reference.

We also keep Fig. 4D as it allows for the analyses of all the attacks together by looking at the predator and prey perspective simultaneously, and have a different methodology of the analysis compared to Figs. 4A-C, as we have detailed in the Supplementary Material lines 976-990, Table S2 and Fig. S15.

Fig. 4D serves for the comparison of the recovery time across the attacks given optimal prey escape angles, while the main focus of Fig. 4A-C is on finding these optimal flee angles for each attack separately.

Figure R6.

Comment 9. The authors mention that they performed manual annotation, but also that they used a custom matlab code to extract escape angles. I'm not sure of this specific journal, but a comment on availability or "code available upon request" is often standard.

Response: Thank you for paying our attention to this. We have added the comment on the "code availability" to lines 653-653 of the resubmission.

Comment 10. Some of the SI material I found a bit lacking, in specific the examples of attack events and their tracking Fig S2 and S3. First, given that the authors only analyzed 16 attack events with fountain escapes, a video of all these events should be added here.

Response: We have added instances of the zoomed footage of all 14 analysed attack events with "fountain" escape in the revised manuscript: 6 back attacks in Fig. S2, 4 side attacks in Fig. S3, 4 front attacks in Fig. S4 (the ones in Fig. 1D in the main text). The instances were taken at the moment of the lowest convexity of the geometry of the prey school for this particular attack event (see Table S3). We also made all 14 videos covering the whole "fountain" duration available on Zenodo: https://doi.org/10.5281/zenodo.13355844 (also here is a direct download link for reviewers: <https://box.hu-berlin.de/f/725871c8badd4b9da100/?dl=1>) and added a respective note on the "data availability" to lines 650-652 of the revised manuscript.

Additionally, we corrected in the main text, line 172 the number of analysed attacks from $n=16$ to $n=14$, as indicated in Fig. 1D.

Comment 11. Addionatily, for the figure itself, a grid of rows representing each attack, and columns representing the "start time, end time, lowest polarization and recovery time", should be added to each example shown.

Response: Thank you for the suggestion. We have added Table S3 in the revised Supplementary Material, where rows represent each attack event and columns state for the attack type, attack id (see also Fig. S7), video id, lowest polarisation value Φ , iteration (1 iter = 0.01 s) at which

achieved the lowest polarisation, lowest prey school convexity value C , iteration at which achieved the lowest convexity, start time of the “fountain effect” t_{start}^{fnt} , end time of the “fountain effect” t_{end}^{fnt} , time shift (in iterations) introduced for Fig. 1D of the main text, recovery time τ .

Below we attach a snapshot of the table from the revised manuscript for the reference.

attack type	id	video	min Φ	iter of min Φ	min C	iter of min C	t_{start}^{fnt}	t_{end}^{fnt}	time shift	recovery time τ
front	23	DJI_3_f3943	0.195	10	0.799	8	2	6	2	10
	25	DJI_3_f4571	0.033	12	0.91	6	3	9	0	8
	15	DJI_2_f4782	0.305	10	0.757	6	1	7	2	7
	20	DJI_3_f10838	0.42	16	0.651	5	1	7	2	11
back	3	DJI_1_f1519	0.193	11	0.666	15	8	14	0	8
	7	DJI_1_f4587	0.411	6	0.737	8	2	8	5	11
	13	DJI_2_f3297	0.306	8	0.502	11	5	11	3	8
	19	DJI_3_f10577	0.048	7	0.739	9	4	9	4	12
	27	DJI_3_f7153	0.165	7	0.726	10	4	9	4	x
	29	DJI_3_f8437	0.29	11	0.704	13	9	13	0	5
side	0	DJI_1_f1355	0.57	5	0.82	9	4	8	0	4
	22	DJI_3_f19240	0.284	5	0.699	9	3	8	0	3
	10	DJI_2_f2586	0.496	4	0.666	7	1	8	1	3
	16	DJI_2_f4878	0.479	4	0.703	6	1	6	1	7

Table S3. Metadata on 14 analysed attack events with “fountain” escapes, describing the lowest polarisation value Φ with respective time (iter of min Φ), the lowest convexity value C of the prey school during the fountain manoeuvre with respective time (iter of min C), the start and the end of the “fountain effect”, time shift introduced for Fig. 1D in the main text, and prey recovery time τ . To align the instances within the attack types by the lowest polarisation value (as in Fig. 1D of the main text), one has to add “time shift” to t_{start}^{fnt} , t_{end}^{fnt} , iter of min Φ and C , where 1 iter equals 0.01 s. The order of the attack IDs corresponds to the lexicographical order of the footage instances in Figs. S2-S4.

Comment 12. Arrow heads representing fish should have the same size, since they don’t represent speed, and only their stem varies its length (given by the manual annotation of head vs tail).

Response: We have adjusted the size of the arrowheads in Figs. S2-S4 of the revised manuscript such that they are consistent within each instance. For plotting the arrows, we used the Python matplotlib quiver function, which originally scales the arrowhead size relative to the arrow length. To prevent this, we normalized the arrow lengths within each instance, ensuring uniform arrow sizes regardless of the annotated positions of the sardines’ heads and tails. The original manually annotated head and tail positions are now marked with orange and yellow dots, respectively.

Comments from Reviewer #3

Comment 1. Figure 1: For figure 1B, consider adding the total number of attacks in the legend (N=104) to avoid confusion with subsequent analyses. Similarly, add the number of attacks in Figure 1C, as the total number of videos included in following analysis is reduced. I noticed the information is included in the caption but I would still recommend adding it to the figures.

Response: We noticed that Reviewer 2 also raised the same point. We have addressed it in our response to Reviewer 2, Comment 3.

Comment 2. Figure 1D: The figure is informative, but I suggest adding information about the shaded bar. To illustrate the variation in start/end times of different trials, consider creating a separate boxplot with all the data points, or adding an error bar around the shaded bar (the extra figure could be in the supplementary material). It would be helpful to know the number of frames each line was shifted after aligning to the minimum polarization value. Also, always for figure 1D, I would recommend making the lines of each encounter (grey lines) more visible.

Response: We noticed that Reviewer 2 also raised the same point. We have addressed it in our response to Reviewer 2, Comment 1. Namely, we have added standard deviation bars to the start/end times of the fountain evasion in Fig. 1D of the main text and made the lines of each encounter more visible by colouring in the darker shade of grey. We reported the number of frames each line was shifted to align with the minimum polarisation value in the revised Supplementary Material in Table S3 as “time shift”.

We also attach below Fig. R7A, which illustrates the variations in the start/end times of different trials. However, we do not find it in general informative as the start/end times depend on the random cuts of the video events from the whole footage and, hence, do not hold scientific information. Therefore, we computed the duration of the fountain manoeuvre which is independent of this in Fig. R7B and included it in the revised Supplementary Material as Fig. S16.

Figure R7. (A) Time windows of the fountain manoeuvre $[t_{start}^{fnt}, t_{end}^{fnt}]$ depending on the attack direction. The error bars depict the corresponding mean and standard deviation values of t_{start}^{fnt} and t_{end}^{fnt} . (B) Duration of the “fountain effect” depending on the attack direction (front, back, and side). According to the Kruskal-Wallis test followed by the Tukey test for multiple pairwise comparisons, the results are non-significantly different (ns).

Comment 3. Time Interval Between Attacks: Are there instances where the same predator attacks more than once within a short time interval? If so, what is the average time interval between attacks by the same individual? The supplementary material mentions one case where the fish school did not recover before the next attack. Does this occur in other attacks, even if they are not included in the analysis? Can any information be extracted about the attack strategy, whether it is the first attack or a subsequent one by the same predator (e.g. side/front/back)?

Response: Thank you for your questions. There are indeed times when the same predator attacks more than once within a short time interval. The mean time between attacks by the same individual is 4.03 seconds (SD=3.4). This is slightly shorter than the mean time between attacks by two different predators (4.62 seconds, SD = 4.23). The time between attacks can range from very short (e.g. less than a second) to up to 30 seconds.

There are times when the school does not fully recover before the next attack, although this is extremely rare as the recovery time of the school was less than a second (0.1 second, Fig 1E). Because of this speed, it is unlikely the same predator can turn around fast enough to attack the school before it has reformed. It more likely that this happens only when a different predator attacks very close in time to the first predator (but this comes with a real risk of collision). However, we did not collect the data to specifically explore the relative frequency of these events nor whether they were related to specific attack angles or attacks by the same or different predators.

Comment 4. Although it may not always be possible due to image quality, is it sometimes possible to determine if a hunt attempt was successful?

Response: It is sometimes possible to determine if an attack was successful or not. We found 25 fountains where we could clearly see either prey contact or a capture. Unfortunately, it is not possible to say that the remaining 42 fountain attacks we witnessed (N=67 fountains in total) were successful or unsuccessful due to poor image quality from splashes or other obfuscations. Of these

25 contacts/ captures 15 were from "side" attacks, 7 were from "back" attacks, and 3 were from "front" attacks.

Comment 5. Figure 2: Considering that Figure 1 reports the recovery time and Figure 2C reports the distance to a predator, could you add to Figure 2 the equivalent of Figure 4D but with empirical data instead of simulation data? I find Figure 4D very interesting for understanding the optimal strategies of both predator and preys and I would expect a similar pattern for empirical data.

Response: Unfortunately, it is not possible to recreate Figure 4D with the empirical data instead of the simulation data. We wanted to do this for the initial submission, however, it was simply not possible to extract these types of data from the video with a level of accuracy we would be confident reporting on. Although we were able to track a sub-sample of the prey fish (N=20) during the fountain escape to characterise the prey's velocity orientations, we could not visualise every prey fish in the school throughout the whole attack sequence. Therefore, we don't believe we can estimate the distance of all prey (as it is in the simulation) or of the closest prey individuals to the predator in the empirical data. On the other hand, this highlights the power of agent-based modelling, as it allows us to gain insights in the absence of complete empirical data.

Comment 6. Figure 3: I would suggest removing this figure and moving it to the supplementary material. I find it hard to understand due to lacking figure labels (e.g., y-label of plots A and C), making it difficult to interpret. Moreover, the figure is mentioned only once in the main text, so it might be better suited for the supplementary material. Alternatively, you could improve it by fixing the figure axes labels and descriptions in the main text. I am unsure if I understand what plots A and C are illustrating and which data is selected for the plot.

Response: We noticed that Reviewer 2 also raised the same point. We have addressed it in our response to Reviewer 2, Comment 5.

Comment 7. Figure 4: I would adjust both x and y axes to use the same range for all figures A-D. This way, I find that it could possibly be easier to compare the recovery times associated with each attack type front, back, side.

Response: We noticed that Reviewer 2 also raised the same point. We have addressed it in our response to Reviewer 2, Comment 8.

Comment 8. Figure S1: 8 predators were subsampled with 4 attacks each. I suggest highlighting in this figure the 16 attacks (if present) also analyzed in the main text. Are these 8 predators also part of the 11 predators of Figure 1D? (I am referring to information written in the second paragraph of the supplementary material). Why were some predators excluded from the ANOVA test?

Response: Out of the 16 attacks analysed in the main text, only 4 are also present in Fig. S1. We highlighted these attacks by annotating them with their respective fountain ID numbers, as listed in Table S3, and by using colours corresponding to their attack direction (blue for back attacks, green for side attacks). Among the 8 predators shown in Fig. S1, only 4 (I, AA, C, LL) are part of the 11 predators featured in Fig. 1D. These 4 predators are now highlighted by annotating their attacks in Fig. S1 of the revised manuscript. We have specified this in the revised Supplementary Material, lines 893-894, and in the caption of Fig. S1.

As we mentioned in the initial submission, lines 888-891 of the revised Supplementary Material, out of the 23 predators that attacked at least twice, we excluded those with fewer than 4 attacks, as we considered 2 or 3 observations to be insufficient for a robust ANOVA test.

Comment 9. Figure S2: Given that Figure 1D includes 16 attacks, would it be possible to include one screenshot per attack as supplementary material? Alternatively, may be better to explain how the 6 instances were selected. What is the information contained in the size of the yellow arrow? If it is not relevant, consider keeping the size equal for each individual. Also, how was the frame of the instance decided? Is it an analogous moment across attacks? For instance, considering all trials were shifted based on the moment of lowest polarization of the group, could the instances in Figure S2 all be from a similar time in the attack?

Response: We noticed that Reviewer 2 also raised the same point. We have addressed it in our response to Reviewer 2, Comment 12. Specifically, we have included one screenshot per each of the 14 analysed attacks in the revised Supplementary Material as Figs. S2-S4. We have adjusted the size of the arrowheads to keep them uniform within each screenshot as it does not contain any information. The length of the arrows was defined by the annotated head and tail of the fish, but in the revised manuscript we keep it normalised within a screenshot to preserve equal size of arrowheads. The frame of the instance corresponds to the analogous moment across attacks which is the moment of the lowest convexity of the geometry of the prey school in that particular instance throughout the “fountain effect”.

Comment 10. Video material: Could you share the videos of the selected attacks along with the publication? This is not necessary but could be an interesting addition.

Response: We noticed that Reviewer 2 also raised the same point. We have addressed it in our response to Reviewer 2, Comment 10.

References

1. P. P. Klamser and P. Romanczuk. Collective predator evasion: Putting the criticality hypothesis to the test. *PLOS Computational Biology*, 17(3):1–21, 03 2021. doi:10.1371/journal.738.pcbi.1008832
2. I. Karamouzas, B. Skinner, and S. J. Guy. Universal power law governing pedestrian interactions. *Phys. Rev. Lett.*, 113:238701, Dec 2014. doi:10.1103/PhysRevLett.113.238701
3. P. Romanczuk, I. D. Couzin, and L. Schimansky-Geier. Collective motion due to individual escape and pursuit response. *Phys. Rev. Lett.*, 102:010602, Jan 2009. doi:78910.1103/PhysRevLett.102.010602
4. S. J. G. Hall, C. S. Wardle, and D. N. MacLennan. Predator evasion in a fish school: test of a model for the fountain effect. *Marine Biology*, 91:143–148, 1986